# BeBold: Exploration Beyond the Boundary of Explored Regions

## Abstract

Efficient exploration under sparse rewards remains a key challenge in deep reinforcement learning. To guide exploration, previous work makes extensive use of intrinsic reward (IR). There are many heuristics for IR, including visitation counts, curiosity, and state-difference. In this paper, we analyze the pros and cons of each method and propose the *regulated difference of inverse visitation counts* as a simple but effective criterion for IR. The criterion helps the agent explore **Be**yond the **Bo**undary of exp**lo**re**d** regions and mitigates common issues in count-based methods, such as *short-sightedness* and *detachment*. The resulting method, **BeBold**, solves the 12 most challenging procedurally-generated tasks in MiniGrid with just 120M environment steps, without any curriculum learning. In comparison, previous SoTA only solves 50% of the tasks. BeBold also achieves SoTA on multiple tasks in NetHack, a popular rogue-like game that contains more challenging procedurally-generated environments.

## 1 Introduction

Deep reinforcement learning (RL) has experienced significant progress over the last several years, with impressive performance in games like Atari (Mnih et al., 2015; Badia et al., 2020a), Star-Craft (Vinyals et al., 2019) and Chess (Silver et al., 2016; 2017; 2018). However, most work requires either a manually-designed dense reward (Brockman et al., 2016) or a perfect environment model (Silver et al., 2017; Moravčík et al., 2017). This is impractical for real-world settings, where the reward is sparse; in fact, the proper reward function for a task is often even unknown due to lack of domain knowledge. Random exploration (e.g., $\epsilon$-greedy) in these environments is often insufficient and leads to poor performance (Bellemare et al., 2016).

Recent approaches have proposed to use **intrinsic rewards** (IR) (Schmidhuber, 2010) to motivate agents for exploration before any extrinsic rewards are obtained. Various criteria have been proposed, including *curiosity/surprise*-driven (Pathak et al., 2017), *count*-based (Bellemare et al., 2016; Burda et al., 2018a;b; Ostrovski et al., 2017; Badia et al., 2020b), and *state-diff* approaches (Zhang et al., 2019; Marino et al., 2019).

Each approach has its upsides and downsides: Curiosity-driven approaches look for prediction errors in the learned dynamics model and may be misled by the noisy TV (Burda et al., 2018b) problem, where environment dynamics are inherently stochastic. Count-based approaches favor novel states in the environment but suffer from *detachment* and *derailment* (Ecoffet et al., 2019), in which the agent gets trapped into one (long) corridor and fails to try other choices. Count-based approaches are also *short-sighted*: the agent often settles in local minima, sometimes oscillating between two states that alternately feature lower visitation counts (Burda et al., 2018b). Finally, state-diff approaches offer rewards if, for each trajectory, representations of consecutive states differ significantly. While these approaches consider the entire trajectory of the agent rather than a local state, it is asymptotically inconsistent: the intrinsic reward remains positive when the visitation counts approach infinity. As a result, the final policy does not necessarily maximize the cumulative extrinsic reward.

In this paper, we propose a novel exploration criterion that combines count-based and state-diff approaches: instead of using the difference of state representations, we use the *regulated difference of inverse visitation counts* of consecutive states in a trajectory. The inverse visitation count is approximated by Random Network Distillation (Burda et al., 2018b). Our IR provides two benefits: (1) This addresses asymptotic inconsistency in the state-diff, since the inverse visitation count vanishes with sufficient explorations. (2) Our IR is large at the end of a trajectory and at the *boundary* between the explored and the unexplored regions (Fig. 1). This motivates the agent to move **Be**yond

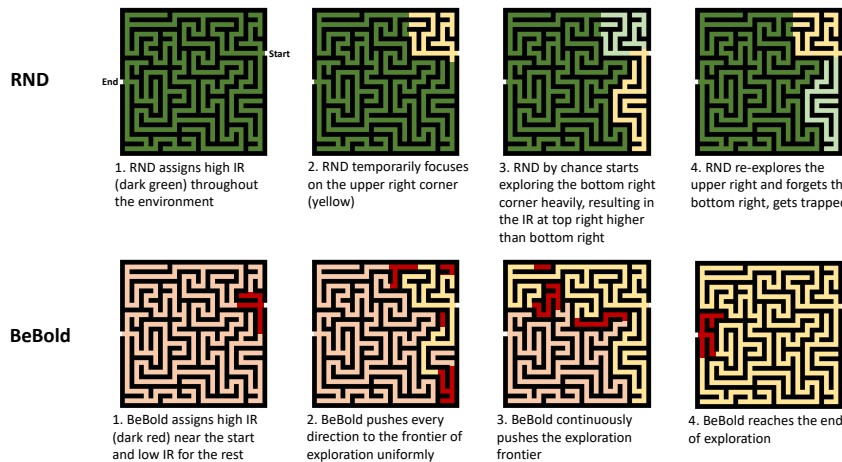

Figure 1: A Hypothetical Demonstration of how exploration is done in BeBold versus Random Network Distillation (Burda et al., 2018b), in terms of distribution of intrinsic rewards (IR). BeBold reaches the goal by continuously pushing the frontier of exploration while RND got trapped. Note that IR is defined differently in RND ($1/N(\mathbf{s}_t)$) versus BeBold ($\max(1/N(\mathbf{s}_{t+1}) - 1/N(\mathbf{s}_t), 0)$, See Eqn. 3), and different color is used.

the **Bo**undary of the exp**lo**re**d** regions and step into the unknown, mitigating the short-sighted issue in count-based approaches.

Following this simple criterion, we propose a novel algorithm **BeBold** and evaluate it on two very challenging procedurally-generated (PG) environments: MiniGrid (Chevalier-Boisvert et al., 2018) and NetHack (Küttler et al., 2020). MiniGrid is a popular benchmark for evaluating exploration algorithms (Raileanu and Rocktäschel, 2020; Campero et al., 2020; Goyal et al., 2019) and NetHack is a much more realistic environment with complex goals and skills. BeBold manages to solve the 12 most challenging environments in MiniGrid within 120M environment steps, without curriculum learning. In contrast, (Campero et al., 2020) solves 50% of the tasks, which were categorized as "easy" and "medium", by training a separate goal-generating teacher network in 500M steps. In NetHack, a more challenging procedurally-generated environment, BeBold also outperforms all baselines with a significant margin on various tasks. In addition, we analyze BeBold extensively in MiniGrid. The quantitative results show that BeBold largely mitigates the *detachment* problem, with a much simpler design than Go-Explore (Ecoffet et al., 2020) which contains multiple hand-tune stages and hyper-parameters.

**Most Related Works**. RIDE (Raileanu and Rocktäschel, 2020) also combines multiple criteria together. RIDE learns the state representation with curiosity-driven approaches, and then uses the difference of learned representation along a trajectory as the reward, weighted by pseudo counts of the state. However, as a two-stage approach, RIDE heavily relies on the quality of generalization of the learned representation on novel states. As a result, BeBold shows substantially better performance in the same procedurally-generated environments.

Go-Explore (Ecoffet et al., 2020) stores many visited states (including boundaries), reaches these states without exploration, and explores from them. BeBold focuses on boundaries, perform exploration without human-designed cell representation (e.g., image downsampling) and is end-to-end.

Frontier-based exploration (Yamauchi, 1997; 1998; Topiwala et al., 2018) is used to help specific robots explore the map by maximizing the information gain. The "frontier" is defined as the 2D spatial regions out of the explored parts. No automatic policy optimization with deep models is used. In contrast, BeBold can be applied to more general partial observable MDPs with deep policies.

## 2 BACKGROUND

Following single agent Markov Decision Process (MDP), we define a state space $S$, an action space $A$, and a (non-deterministic) transition function $\mathcal{T} : S \times A \to P(S)$ where $P(S)$ is the probability of next state given the current state and action. The goal is to maximize the expected reward $R = \mathbb{E}[\sum_{k=0}^{T} \gamma^k r_{t+k=1}]$ where $r_t$ is the reward, $\gamma$ is the discount factor, and the expectation is taken w.r.t. the policy $\pi$ and MDP transition $P(S)$. In this paper, the total reward received at time step $t$ is given by $r_t = r_t^e + \alpha r_t^i$, where $r_t^e$ is the extrinsic reward given by the environment, $r_t^i$ is the intrinsic reward from the exploration criterion, and $\alpha$ is a scaling hyperparameter.

## 3  EXPLORATION BEYOND THE BOUNDARY

Our new exploration criterion combines both counting-based and state-diff-based criteria. Our criterion doesn't suffer from short-sightedness and is asymptomatically consistent. We'll first introduce BeBold and then analyse the advantages of BeBold over existing criteria in Sec. 4.

**Exploration Beyond the Boundary**. BeBold gives intrinsic reward (IR) to the agent when it explores beyond the boundary of explored regions, i.e., along a trajectory, the previous state $\mathbf{s}_t$ has been sufficiently explored but $\mathbf{s}_{t+1}$ is new:

$$r^i(\mathbf{s}_t, \mathbf{a}_t, \mathbf{s}_{t+1}) = \max\left(\frac{1}{N(\mathbf{s}_{t+1})} - \frac{1}{N(\mathbf{s}_t)}, 0\right), \tag{1}$$

Here $N$ is the visitation counts. We clip the IR here because we don't want to give a negative IR to the agent if it transits back from a novel state to a familiar state. From the equation, only crossing the frontier matters to the intrinsic reward; if both $N(\mathbf{s}_t)$ and $N(\mathbf{s}_{t+1})$ are high or low, their difference would be small. As we will show in Sec. 4, for each trajectory going towards the frontier/boundary, BeBold assigns an approximately equal IR, regardless of their length. As a result, the agent will continue pushing the frontier of exploration in a much more uniform manner than RND and won't suffer from short-sightedness. This motivates the agent to explore different trajectories uniformly. Also Eq. 1 is asymptotically consistent as $r^i \to 0$ when $N \to \infty$.

Like RIDE (Raileanu and Rocktäschel, 2020), in our implementation, partial observation $\mathbf{o}_t$ are used instead of the real state $\mathbf{s}_t$, when $\mathbf{s}_t$ is not available.

**Episodic Restriction on Intrinsic Reward (ERIR).** In many environments where the state transition is reversible, simply using intrinsic reward to guide exploration would result in the agent going back and forth between novel states $\mathbf{s}_{t+1}$ and their previous states $\mathbf{s}_t$. RIDE (Raileanu and Rocktäschel, 2020) avoids this by scaling the intrinsic reward $\mathbf{r}(\mathbf{s})$ by the inverse of the state visitation counts. BeBold puts a more aggressive restriction: the agent is only rewarded when it visits the state $\mathbf{s}$ for the first time in an episode. Thus, the intrinsic reward of BeBold becomes:

$$r^i(\mathbf{s}_t, \mathbf{a}_t, \mathbf{s}_{t+1}) = \max\left(\frac{1}{N(\mathbf{s}_{t+1})} - \frac{1}{N(\mathbf{s}_t)}, 0\right) * \mathbb{1}\{N_e(\mathbf{s}_{t+1}) = 1\} \tag{2}$$

$N_e$ here stands for episodic state count and is reset every episode. In contrast, the visitation count $N$ is a life-long memory bank counting state visitation across all of training.

**Inverse visitation counts as prediction difference.** We use the difference between a teacher $\phi$ and a student network $\phi'$ to approximate visitation counts: $N(\mathbf{s}_{t+1}) \approx \frac{1}{||\phi(\boldsymbol{o}_{t+1}) - \phi'(\boldsymbol{o}_{t+1})||_2}$, here $\boldsymbol{o}_{t+1}$ is the observation of the agent in state $\mathbf{s}_{t+1}$. This yields the following implementation of BeBold:

$$r^i(\mathbf{s}_t, \mathbf{a}_t, \mathbf{s}_{t+1}) = \max(||\phi(\boldsymbol{o}_{t+1}) - \phi'(\boldsymbol{o}_{t+1})||_2 - ||\phi(\boldsymbol{o}_t) - \phi'(\boldsymbol{o}_t)||_2, 0) * \mathbb{1}\{N_e(\boldsymbol{o}_{t+1}) = 1\}) \tag{3}$$

**Shared visitation counts $N(\mathbf{s}_t)$ in the training of Procedurally-Generated (PG) Environments.** During training, the environment changes constantly (e.g., blue keys becomes red), while the semantic links of these objects remain the same. We use a shared RND ($\phi$, $\phi'$) across different PG environments, and treat these semantically similar states as new without using domain knowledge (e.g., image downsampling like in Go-Explore (Ecoffet et al., 2019)). Partial observability and generalization of neural network $\phi$ handles these differences and leads to count-sharing. For episodic count $N_e(\boldsymbol{o}_{t+1})$, since it is not shared across episodes (and environments), we use a hash table.

## 4  CONCEPTUAL ADVANTAGES OF BEBOLD OVER EXISTING CRITERIA

**Short-sightedness and Detachment**. One issue in the count-based approach is its short-sightedness. Let's assume in a simple environment, there are $M$ corridors $\{\tau_j\}_{j=1}^M$ starting at $\mathbf{s}_0$ and extending to different parts of the environment. The corridor $\tau_j$ has a length of $T_j$. The agent starts at $\mathbf{s}_0$. For each visited state, the agent receives the reward of $\frac{1}{N(\mathbf{s})}$ where $N(\cdot)$ is the visitation count, and learns with $Q$-learning. Then with some calculation (See Appendix), we see that the agent has a strong preference on exploring the longest corridor first (say $\tau_1$), and only after a long period does it start to explore the second longest. This is because the agent initially receives high IR in $\tau_1$ due to its length, which makes the policy $\pi$ visit $\tau_1$ more often, until it depletes the IR in $\tau_1$.

This behavior of "dedication" could lead to serious issues. If $M \geq 3$ and 2 corridors are long enough (say $\tau_1$ and $\tau_2$ are long), then before the agent is able to explore other corridors, its policy $\pi$

has already been trained long enough so that it only remembers how to get into $\tau_1$ and $\tau_2$. When $\tau_1$ has depleted its IR, the agent goes to $\tau_2$ following the policy. After that, the IR in $\tau_1$ *revives* since the visitation counts in $\tau_1$ is now comparable or even smaller than $\tau_2$, which lures the agent to explore $\tau_1$ again following the policy. This leaves other corridors (e.g., $\tau_3$) unexplored for a very long time. Note that using a neural-network-approximated IR (RND) instead of tabular IR could potentially alleviate this issue, but it is often far less than enough in complex environments.

As mentioned in Go-Explore series (Ecoffet et al., 2019; 2020), count-based approaches also suffer from *detachment*: if the agent by chance starts exploring $\tau_2$ after briefly exploring the first few states of $\tau_1$, it would not return and explore $\tau_1$ further since $\tau_1$ is now "shorter" than $\tau_2$ and has lower IR than $\tau_2$ for a long period. Go-Explore tries to resolve this dilemma between "dedication" and "exploration" by using a two-stage approach with many hand-tuned parameters.

In contrast, IR of BeBold depends on the *difference* of the visitation counts along the trajectory, and is insensitive to the *length* of the corridor. This leads to *simultaneous* exploration of multiple corridors and yields a diverse policy $\pi$ (See Sec. 5.2 for empirical evidence). Moreover, the IR focuses on the boundary between explored and unexplored regions, where the two goals (dedication and exploration) align, yielding a much cleaner, one-stage method.

**Asymptotic Inconsistency.** Approaches that define IR as the difference between state representations $\|\psi(\mathbf{s}) - \psi(\mathbf{s}')\|$ ($\psi$ is a learned embedding network) (Zhang et al., 2019; Marino et al., 2019) suffer from asymptotic inconsistency. In other words, their IR does not vanish even after sufficient exploration: $r^i \not\to 0$ when $N \to \infty$. This is because when the embedding network $\psi$ converges after sufficient exploration, the agent can always obtain non-zero IR if a major change in state representation occurs (e.g., opening a door or picking up a key in MiniGrid). Therefore, the learned policy does not maximize the extrinsic reward $r^e$, deviating from the goal of RL. Automatic curriculum approaches (Campero et al., 2020)) have similar issues due to an ever-present IR.

For this, (Zhang et al., 2019) proposes to learn a separate scheduler to switch between intrinsic and extrinsic rewards, and (Raileanu and Rocktäschel, 2020) divides the state representation difference by the square root of visitation counts. In comparison, BeBold does not require any extra stage and is a much simpler solution.

## 5 EXPERIMENTS

We evaluate BeBold on challenging procedurally-generated environment MiniGrid (Chevalier-Boisvert et al., 2018) and the hard-exploration environment NetHack (Küttler et al., 2020). These environments provide a good testbed for exploration in RL since the observations are symbolic rather than raw sensor input (e.g., visual input), which decouples perception from exploration. In MiniGrid, we compare BeBold with RND (Burda et al., 2018b), ICM (Pathak et al., 2017), RIDE (Raileanu and Rocktäschel, 2020) and AMIGo (Campero et al., 2020). We only evaluate AMIGo for 120M steps in our experiments, the algorithm obtains better results when trained for 500M steps as shown in (Campero et al., 2020). For all the other baselines, we follow the exact training paradigm from (Raileanu and Rocktäschel, 2020). Mean and standard deviation across four runs of different seeds are computed. BeBold successfully solves the 12 most challenging environments provided by MiniGrid. By contrast, all the baselines end up with zero reward on half of the environments we tested. In NetHack, BeBold also achieves SoTA results with a large margin over baselines.

### 5.1 MINIGRID ENVIRONMENTS

We mainly use three challenging environments from MiniGird: *Multi-Room* (**MR**), *Key Corridor* (**KC**) and *Obstructed Maze* (**OM**). We use these abbreviations for the remaining of the paper (e.g., `OM2Dlh` stands for ObstructedMaze2Dlh). Fig. 2 shows one example of a rendering on `OMFull` as well as all the environments we tested with their relative difficulty.

In MiniGrid, all the environments are size $N \times N$ ($N$ is environment-specific) where each tile contains an object: wall, door, key, ball, chest. The action space is defined as turn left, turn right, move forward, pick up an object, drop an object, and toggle an object (e.g., open or close a door). **MR** consists of a series of rooms connected by doors and the agent must open the door to get to the next room. Success is achieved when the agent reaches the goal. In **KC**, the agent has to explore the environment to find the key and open the door along the way to achieve success. **OM** is the hardest: the doors are locked, the keys are hidden in boxes, and doors are obstructed by balls.

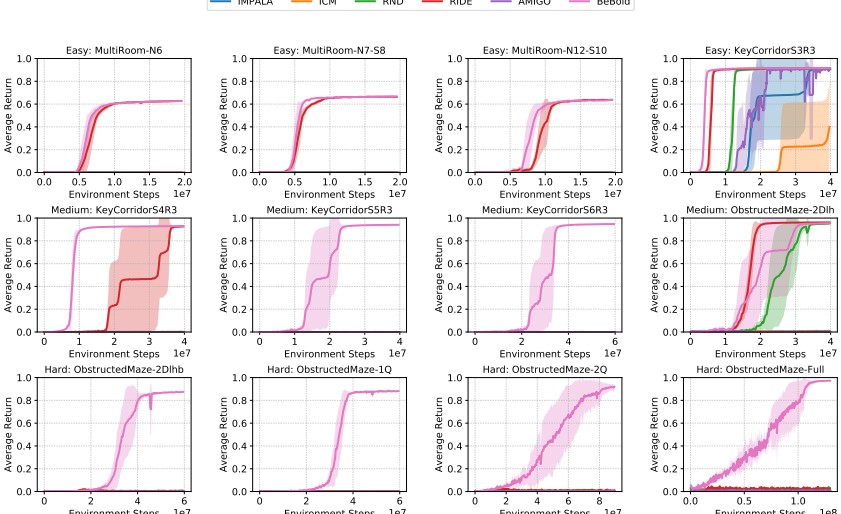

Figure 2: MiniGrid Environments. **Left:** a procedurally-generated `OMFull` environment. **Right:** BeBold solves challenging tasks which previous approaches cannot solve. Note that we evaluate all methods for 120M steps. AMIGo gets better results when trained for 500M steps as shown in (Campero et al., 2020).

Figure 3: Results for various hard exploration environments from MiniGrid. BeBold successfully solves all the environments while all other baselines only manage to solve two to three relatively easy ones.

**Results**. We test BeBold on all environments from MiniGrid. BeBold manages to solve the 12 most challenging environments. By contrast, all baselines solve only up to medium-level tasks and fail to make any progress on more difficult ones. Note that some medium-level tasks we define here are categorized as hard tasks in RIDE and AMIGo (e.g., `KCS4R3` is labeled as "KCHard" and `KCS5R3` is labeled as "KCHarder"). Fig. 3 shows the results of our experiments. Half of the environments (e.g., `KCS6R3`, `OM1Q`) are extremely hard and all the baselines fail. In contrast, BeBold easily solves all such environments listed above without any curriculum learning. We also provide the final testing performance for BeBold in Tab. 1. The results is averaged across 4 seeds and 32 random initialized environments.

Multi Room environments are relatively easy in MiniGrid. However, all the baselines except RIDE fail. As we increase the room size and number (e.g., `MRN12S10`), BeBold can achieve the goal quicker than RIDE. Our method easily solves these environments within 20M environment steps.

On Key Corridor environments, RND, AMIGo, RIDE, IMPALA and BeBold successfully solves `KCS3R3` while ICM makes reasonable progress. However, when we increase the room size (e.g., `KCS5R3`), none of the baseline methods work. BeBold manages to solve these environments in 40M environment steps. The agent demonstrates the ability to explore the room and finds the corresponding key to open the door in a randomized, procedurally-generated environment.

Obstructed Maze environments are also difficult. As shown in Fig. 3, RIDE and RND manage to solve the easiest task `OM2D1h` which doesn't contain any obstructions. In contrast, BeBold not only rapidly solves `OM2D1h`, but also solves four more challenging environments including `OMFull`. These environments have obstructions blocking the door (as shown in Fig. 2) and are much larger in size than `OM2D1h`. In these environments, our agent learns to move the obstruction away from the door to open the door and enter the next room. This "skill" is hard to learn since there is no extrinsic reward assigned to moving the obstruction. However, learning the skill is critical to achieve the goal.

Table 1: Final testing performance for BeBold and all baselines.

|  | MRN6 | MRN7S8 | MRN12S10 | KCS3R3 | KCS4R3 | KCS5R3 |
|---|---|---|---|---|---|---|
| ICM | $0.00 \pm 0.0$ | $0.00 \pm 0.0$ | $0.00 \pm 0.0$ | $0.45 \pm 0.052$ | $0.00 \pm 0.0$ | $0.00 \pm 0.0$ |
| RIDE | $\mathbf{0.65} \pm 0.005$ | $0.67 \pm 0.001$ | $0.65 \pm 0.002$ | $0.91 \pm 0.003$ | $0.93 \pm 0.002$ | $0.00 \pm 0.0$ |
| RND | $0.00 \pm 0.0$ | $0.00 \pm 0.0$ | $0.00 \pm 0.0$ | $0.91 \pm 0.003$ | $0.00 \pm 0.0$ | $0.00 \pm 0.0$ |
| IMPALA | $0.00 \pm 0.0$ | $0.00 \pm 0.0$ | $0.00 \pm 0.0$ | $0.91 \pm 0.004$ | $0.00 \pm 0.0$ | $0.00 \pm 0.0$ |
| AMIGO | $0.00 \pm 0.0$ | $0.00 \pm 0.0$ | $0.00 \pm 0.0$ | $0.89 \pm 0.005$ | $0.00 \pm 0.0$ | $0.00 \pm 0.0$ |
| BeBold | $0.64 \pm 0.003$ | $\mathbf{0.67} \pm 0.001$ | $\mathbf{0.65} \pm 0.002$ | $\mathbf{0.92} \pm 0.003$ | $\mathbf{0.93} \pm 0.003$ | $\mathbf{0.94} \pm 0.001$ |

|  | KCS6R3 | OM2Dlh | OM2Dlhb | OM1Q | OM2Q | OMFULL |
|---|---|---|---|---|---|---|
| ICM | $0.00 \pm 0.0$ | $0.00 \pm 0.0$ | $0.00 \pm 0.0$ | $0.00 \pm 0.0$ | $0.00 \pm 0.0$ | $0.00 \pm 0.0$ |
| RIDE | $0.00 \pm 0.0$ | $0.95 \pm 0.015$ | $0.00 \pm 0.0$ | $0.00 \pm 0.0$ | $0.00 \pm 0.0$ | $0.00 \pm 0.0$ |
| RND | $0.00 \pm 0.0$ | $0.95 \pm 0.0066$ | $0.00 \pm 0.0$ | $0.00 \pm 0.0$ | $0.00 \pm 0.0$ | $0.00 \pm 0.0$ |
| IMPALA | $0.00 \pm 0.0$ | $0.00 \pm 0.0$ | $0.00 \pm 0.0$ | $0.00 \pm 0.0$ | $0.00 \pm 0.0$ | $0.00 \pm 0.0$ |
| AMIGO | $0.00 \pm 0.0$ | $0.00 \pm 0.0$ | $0.00 \pm 0.0$ | $0.00 \pm 0.0$ | $0.00 \pm 0.0$ | $0.00 \pm 0.0$ |
| BeBold | $\mathbf{0.94} \pm 0.017$ | $\mathbf{0.96} \pm 0.005$ | $\mathbf{0.89} \pm 0.063$ | $\mathbf{0.88} \pm 0.067$ | $\mathbf{0.93} \pm 0.028$ | $\mathbf{0.96} \pm 0.058$ |

Figure 4: Normalized visitation counts $N(\mathbf{s}_t)/Z$ ($Z$ is a normalization constant) for the location of agents. BeBold successfully explores all rooms at $4.6$M steps while RND gets stuck in the fifth room at $9.8$M steps.

## 5.2 Analysis of Intrinsic Reward using Pure Exploration

We analyze how BeBold mitigates the short-sightedness issue by only using IR to guide exploration.

**Shorted-Sighted Problem in Long-Corridor Environment.** To verify Sec. 4, we design a toy environment with four disconnected corridors with length 40, 10, 30, and 10 respectively starting from the same state. In this example, there is no extrinsic reward and the exploration of the agent is guided by IR. We combine Q-learning with tabular IR (count-based and BeBold tabular) and neural-network-approximated IR (RND and BeBold) respectively for this experiment. We remove clipping from BeBold for a fair comparison. Tab. 2 shows the visitation counts across 4 runs w.r.t. each corridor after 600 episodes of training. It is clear that BeBold tabular explores each corridor in a much more uniform manner. On the other hand, count-based approaches are greatly affected by short-sightedness and focus only on two out of four corridors. BeBold also shows much more stable performance across runs as the standard deviation is much lower comparing with RND. Note that comparing to the analysis in Sec. 4, in practical experiments, we perceive that the preference of the corridors for count-based methods can be arbitrary because of the random initialization of the $Q$-network.

**Visitation Counts Analysis in MiniGrid.** To study how different intrinsic rewards can affect the exploration of the agent, we test BeBold and RND in a fixed (instead of procedurally-generated for simplicity) MRN7S8 environment. The environment contains 7 rooms connected by doors. To be a successful exploration strategy, the agent should explore all states and give all states equal amount of exploration. We define two metrics to measure the effectiveness of an exploration strategy: (1) visitation counts at every state over training $N(\mathbf{s})$, and (2) entropy of visitation counts *in each room*: $\mathcal{H}(\rho'(\mathbf{s}))$ where $\rho'(\mathbf{s}) = \frac{N(\mathbf{s})}{\sum_{\mathbf{s} \in \mathcal{S}_r} N(\mathbf{s})}$. We do not calculate entropy across all states, because the agent always starts in the first room and may not visit the last one as frequently. As a result, the visitation counts for states in the first room will be several magnitudes larger than the last room.

Fig. 4 shows the heatmap of normalizd visitation counts $N(\mathbf{s}_t)/Z$, where $Z$ is the normalization constant. At first, RND enters the second room faster than BeBold. However, BeBold consistently makes progress by pushing the *frontier* of exploration and discovers all the rooms in 5M steps, while RND gets stuck in the fifth room even trained with 10M steps.

In Tab. 3, the entropy of distribution in each room $\mathcal{H}(\rho'(\mathbf{s}))$ for BeBold is larger than that of RND. This suggests that BeBold encourages the agent to explore the state in a much more uniform manner.

Table 2: Visitation counts for the toy corridor environment after 3K episodes. BeBold explores corridors more uniformly than count-based approaches.

|  | C1 | C2 | C3 | C4 | Entropy |
|---|---|---|---|---|---|
| Length | 40 | 10 | 30 | 10 | – |
| Count-Based | 66K ± 28K | 8K ± 8K | 23K ± 35K | 13K ± 18K | 1.06 ± 0.39 |
| BeBold Tabular | 26K ± 2K | 28K ± 8K | 25K ± 6K | 29K ± 9K | **1.97 ± 0.02** |
| RND | 0.2K ± 0.2K | 70K ± 53K | 0.2K ± 0.07K | 26K ± 44K | 0.24 ± 0.28 |
| BeBold | 27K ± 6K | 23K ± 3K | 31K ± 12K | 26K ± 8K | **1.96 ± 0.05** |

Table 3: Entropy of the visitation counts of each room. Such state distribution of Be-Bold is much more uniform than RND.

|  | 0.2M | 0.5M | 2.0M | 5.0M |
|---|---|---|---|---|
| Room1 | 3.48 / **3.54** | 3.41 / **3.53** | 3.51 / **3.56** | 3.49 / **3.56** |
| Room2 | **2.87** / – | 3.09 / **3.23** | 3.51 / **3.53** | 3.35 / **3.56** |
| Room3 | – / – | – / – | – / **4.02** | 3.42 / **4.01** |
| Room4 | – / – | – / – | – / **2.74** | 2.85 / **2.87** |

\* Results are presented in the order of "RND / BeBold".

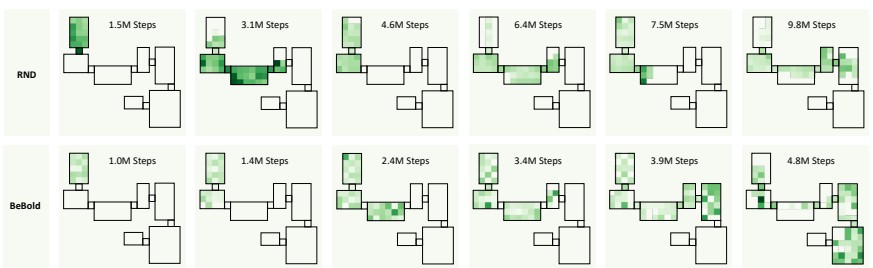

Figure 5: IR heatmaps for the location of agents. BeBold mitigates the short-sighted problem.

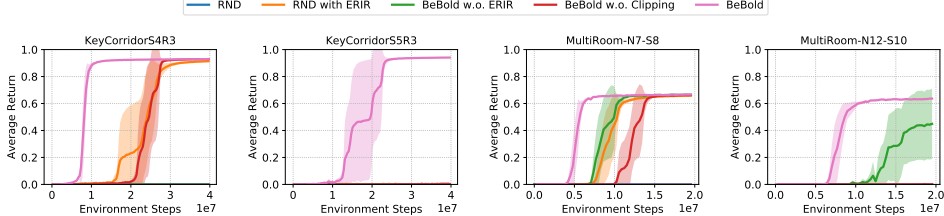

Figure 6: Ablation Study on BeBold comparing with RND with episodic intrinsic reward. BeBold significantly outperforms RND with episodic intrinsic reward on all the environments.

**Intrinsic Reward Heatmap Analysis in MiniGrid**. We also plot the heatmap of IR at different training steps. We generate the plot by running the policy from different checkpoints for 2K steps and plot the IR associated with each state in the trajectory. States that do not receive IR from the sampled trajectories are left blank. For BeBold, the IR is computed as the difference of inverse visitation counts (approximated by RND) between consecutive states $s_t$ and $s_{t+1}$ in the trajectory. From Fig. 5, we can see that BeBold doesn't suffer from short-sighted problem, as we can clearly see that the areas with high IRs are continuously pushed forward from room1 to room7. This is true of the whole training process. On the contrary, the IR heatmap for RND bounces between two consecutive rooms. This is due to short-sightedness: when exploring the second room, the IR of that room will significantly decrease. Since (a) the policy has assigned the first room non-zero probability and (b) the first room now has lower vistation count, RND will revisit the first room. This continues indefinitely, as the agent oscillates between the two rooms.

## 5.3 ABLATION STUDY

**Episodic Restriction on Intrinsic Reward**. We analyze the importance of each component in BeBold. To illustrate the importance of exploring beyond the boundary, we compare BeBold with a slightly modified RND: RND with ERIR. We only give RND intrinsic reward when it visits a new state for the first time in an episode. We can see in Fig. 6 that although ERIR helps RND to solve `KCS4R3` and `MRN7S8`, without BeBold, the method still fails on more challenging tasks `KCS5R3` and `MRN12S10`. A symmetric experiment of removing ERIR from BeBold is also conducted.

**Clipping in Beyond the Boundary Exploration**. We also study the role of clipping $\max(\cdot, 0)$ in our method. Fig. 6 shows the effect of removing clipping from BeBold. We conclude it is suboptimal to design an IR that incurs negative reward when transitioning from an unfamiliar to a familiar state.

## 5.4 THE NETHACK LEARNING ENVIRONMENT

To evaluate BeBold on a more challenging and realistic environment, we choose the NetHack Learning Environment (Küttler et al., 2020). In the game, the player is assigned a role of hero at the beginning of the game. The player needs to descend over 50 procedurally-generated levels to the bottom and find "Amulet of Yendor" in the dungeon. The procedure can be described as first retrieve

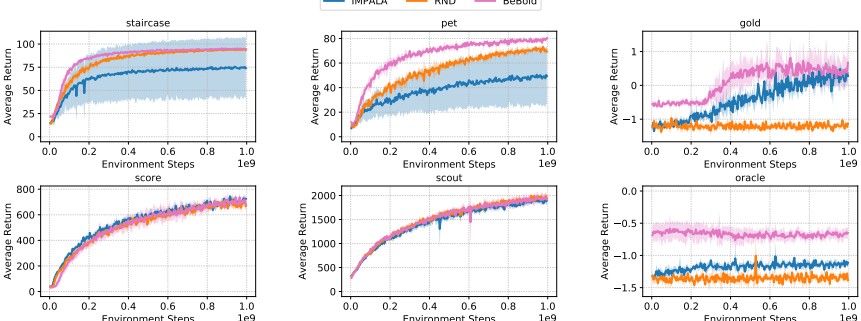

Figure 7: Results for tasks on NetHack. BeBold achieves the SoTA results comparing to RND and IMPALA.

the amulet, then escape the dungeon and finally unlock five challenging levels (the four Elemental Planes and the Astral Plane). We test BeBold on a set of tasks with tractable subgoals in the game: **Staircase**: navigating to a staircase to the next level, **Pet**: reaching a staircase and keeping the pet alive, **Gold**: collecting gold, **Score**: maximizing the score in the game, **Scout**: scouting to explore unseen areas in the environment, and **Oracle**: finding the oracle (an in-game character at level 5-9).

Results in Fig. 7 show that BeBold surpasses RND and IMPALA on all tasks[1]. Especially on **Pet**, BeBold outperforms the other two with a huge margin. This again illustrates the strong performance of BeBold in an environment with huge action spaces and long-horizon reward (several magnitudes longer than StarCraft and Dota2). **Oracle** is the hardest task and no approaches are able to find the oracle and obtain a reward of 1000. BeBold still manages to find a policy with less negative rewards (i.e., penalty of taking actions that do not lead to game advancement, like moving towards a wall). For RIDE, we contacted the authors, who confirmed their method is not functional on NetHack. We further attempted to tune RIDE but to no avail. So we do not report RIDE performance on NetHack.

## 6 RELATED WORK

In addition to the two criteria (*count*-based and *state-diff* based) mentioned above, another stream of defining IRs is *curiosity*-based. The main idea is to encourage agents to explore areas where the prediction of the next state from the current learned dynamical model is wrong. Dynamic-AE (Stadie et al., 2015) computes the distance between the predicted and the real state on the output of an autoencoder, ICM (Pathak et al., 2017) learns the state representation through a forward and inverse model and EMI (Kim et al., 2018) computes the representation through maximizng mutual information $\mathcal{I}([\mathbf{s}, a]; \mathbf{s}')$ and $\mathcal{I}([\mathbf{s}, \mathbf{s}']; a)$.

Another line of research is using information gain to reward the agent. VIME (Houthooft et al., 2016) uses a Bayesian network to measure the uncertainty of the learned model. Later, to reduce computation, a deterministic approach has been adopted (Achiam and Sastry, 2017). Other works also propose to use ensemble of networks for measuring uncertainty (Pathak et al., 2019; Shyam et al., 2019). We can also reward the agent by Empowerment (Klyubin et al., 2005; Gregor et al., 2016; Salge et al., 2014; Mohamed and Rezende, 2015), prioritizing the states that agent can take control through its actions. It is different from state-diff: if $\mathbf{s}_{t+1}$ differs from $\mathbf{s}_t$ but *not* due to agent's choice of actions, then the empowerment at $\mathbf{s}_t$ is zero. Other criteria exist, e.g., diversity (Eysenbach et al., 2018), feature control (Jaderberg et al., 2016; Dilokthanakul et al., 2019) or the KL divergence between current distribution over states and a target distribution of states (Lee et al., 2019).

Outside of intrinsic reward, researchers have proposed to use randomized value functions to encourage exploration (Osband et al., 2016; Hessel et al., 2017; Osband et al., 2019). Adding noise to the network is also shown to be effective (Fortunato et al., 2017; Plappert et al., 2017). There has also been effort putting to either explicitly or implicitly separate exploration and exploitation (Colas et al., 2018; Forestier et al., 2017; Levine et al., 2016). The recently proposed Go-Explore series (Ecoffet et al., 2019; 2020) also fall in this category. We might also set up different goals for exploration (Guo and Brunskill, 2019; Oh et al., 2018; Andrychowicz et al., 2017).

Curriculum learning (Bengio et al., 2009) has also been used to solve hard exploration environments. The curriculum can be explicitly generated by: searching the space (Schmidhuber, 2013), teacher-

---

[1]For **Gold** task, there is a small fix introduced to the environment recently. We benchmark all methods before the fix and will update the results.

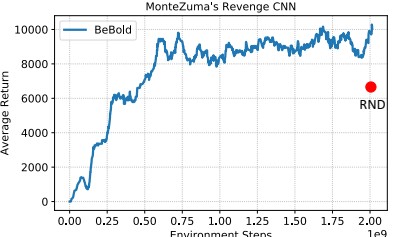 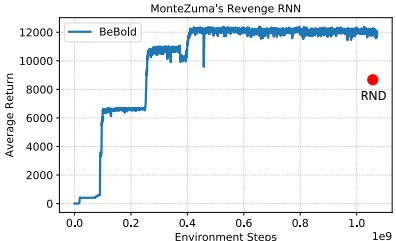

Figure 8: Results for CNN-based and RNN-based model on MonteZuma's Revenge. BeBold achieves good performance.

student setting (Matiisen et al., 2019), increasing distance between the starting point and goal (Jabri et al., 2019) or using a density model to generate a task distribution for the meta learner (Florensa et al., 2017). Our work can also be viewed as an implicit curriculum learning as gradually encourages the agent to expand the area of exploration. However, it never explicitly generates curriculum.

# 7    MONTEZUMA'S REVENGE

We also provide initial result of BeBold on MonteZuma's Revenge. We same paradigm as RND and the same set of hyperparameters except we use 128 parallel environments. In Fig. 8, we can see that using CNN-based model, BeBold achieves approximately 10000 external reward after two Billion frames while the performance reported in RND (Burda et al., 2018b) is around 6700. When using a RNN-baed model, BeBold reached around 13000 external reward in 100K updates while RND only achieves 4400. Please note that these are only initial result and we'll provide the comparison with RND and average return across multiple seeds in the future.

# 8    LIMITATIONS AND FUTURE WORK

**Noisy TV Problem**. One of the limitations on BeBold is the well-known Noisy TV problem. The problem was raised by (Pathak et al., 2017): the agent trained using a count-based IR will get attracted to local sources of entropy in the environment. Thus, it will get high IR due to the randomness in the environment even without making any movements. BeBold suffers from this problem as well since the difference between consecutive states can be caused by the stochasity in the environment. That could be the reason that BeBold doesn't get a good performance on stochastic tasks (e.g., `Dynamic-Obstacles-5x5`). We will leave this problem to future research.

**Hash Table for ERIR**. The ERIR in BeBold adopts a hash table for episodic visitation count. This could potentially have a hard time when applying BeBold in a continuous-space environment (e.g., some robotics tasks). One simple solution is to discretilize the space and still use a hash table for counting. We also leave the more general and elegant fix to this problem to future work.

# 9    CONCLUSION

In this work, we propose a new criterion for intrinsic reward (IR) that encourages exploration beyond the boundary of explored regions using regulated difference of inverse visitation count along a trajectory. Based on this criterion, the proposed algorithm BeBold successfully solves 12 of the most challenging tasks in a procedurally-generated MiniGrid environment. This is a significant improvement over the previous SoTA, overcoming short-sightedness issues that plague count-based exploration. We also evaluate BeBold on NetHack, a much more challenging environment. BeBold outperforms all the baselines by a significant margin. In summary, this simple criterion and the ensuing algorithm demonstrates effectiveness in solving the sparse reward problem in reinforcement learning (RL), opening up new opportunities to many real-world applications.

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

# A  SOME ANALYSIS BETWEEN COUNT-BASED APPROACHES AND OUR APPROACH

Consider two corridors, left and right. The left corridor has length $T_l$ while the right corridor has length $T_r$. Both starts with an initial state $s_0$. When the game starts, the agent was always placed at $s_0$.

For simplicity, we just assume there is only a binary action (left or right) to be chosen at the starting point $s_0$, after that the agent just moves all the way to the end of the corridor and the game restarts. And we set the discount factor $\gamma = 1$.

Using count-based approach, the accumulated intrinsic reward received by the agent if moving to the left corridor for the $i$-th time is $R_l := T_l/i$. This is because for each state, the first time the agent visit it, it gets a reward of 1 and the second time it gets a reward of $1/2$, etc. And this is true for every state in this corridor. Similarly, for $i$-th time moving right, it is $R_r := T_r/i$.

Now let's think how the policy of the agent is trained. In general, the probability to take action left or right is proportional to the accumulated reward. In $Q$ learning it is an exponential moving average of the past reward due to the update rule $Q(s, a) \leftarrow (1 - \alpha)Q(s, a) + \alpha R$ for $\alpha$ typically small (e.g., 0.01). If we think about $Q$ learning starting with $Q(s_0, \text{left}) = Q(s_0, \text{right}) = 0$, and consider the situation when the agent has already done $n_l$ left and $n_r$ right actions, then we have:

$$Q(s_0, \text{left}) = T_l \alpha \sum_{i=1}^{n_l} \frac{(1 - \alpha)^{n_l - i}}{i} \tag{4}$$

Similarly, we have

$$Q(s_0, \text{right}) = T_r \alpha \sum_{i=1}^{n_r} \frac{(1 - \alpha)^{n_r - i}}{i} \tag{5}$$

We could define $\phi(n) := \alpha \sum_{i=1}^{n} \frac{(1-\alpha)^{n-i}}{i}$ and thus $Q(s_0, \text{left}) = T_l \phi(n_r)$ and $Q(s_0, \text{right}) = T_r \phi(n_r)$.

If we treat $\phi(\cdot)$ as a function in the real domain, and assign visiting probability $\pi(a|s_0) \propto Q(s_0, a)$, then the visitation counts, now is denoted as $x_l$ and $x_r$ since they are continuous, for left and right satisfies the following differential equations:

$$\dot{x}_l = \frac{T_l \phi(x_l)}{T_l \phi(x_l) + T_r \phi(x_r)}, \quad \dot{x}_r = \frac{T_r \phi(x_r)}{T_l \phi(x_l) + T_r \phi(x_r)} \tag{6}$$

In the following, we will show that if $T_l \neq T_r$, long corridor will dominate the exploration since it has a lot of rewards until very late. Suppose $x_l(0) = x_r(0) = 1$ and $T_l > T_r$, that is, left side has longer corridor than right side. Then from the equations, very quickly $x_l > x_r$. For $\alpha$ close to 0, we could check that $\phi(x)$ is a monotonously increasing function when $x$ is a small positive number, since the more an agent visits a corridor, the more reward it obtains. So this creates a positive feedback and the growth of $x_l$ dominates, which means that the policy will almost always visit left. This happens until after $x_l$ is large enough and $\phi(x_l)$ starts to decay because of diminishing new rewards and the training discount $\alpha$. The decay is on the order of like $\frac{1}{x_l}$, then the agent will finally start to visit the right side. Indeed, $\pi(\text{right}|s_0) > 50\%$ if $T_r \alpha > T_l/x_l$, or $x_l > \frac{T_l}{T_r} \frac{1}{\alpha}$. In comparison, if we pick sides in a uniform random manner, when $x_l$ is this big, $x_r$ should also be comparable (while here $x_r$ can be as small as 1 or 2).

If an agent have $K$ corridors to pick with $T_1 > T_2 > \ldots > T_K$, things will be similar.

When $T_l = T_r$, the dynamics is perfectly symmetric and the agent visits both left and right with $1/2$ probability. Since this symmetry is quite rare, count-based exploration is often biased towards one or two corridors (See Tbl. 2), BeBold uses difference of inverse visitation counts as intrinsic rewards, and thus balance the exploration.

# B  HYPERPARAMETERS FOR MINIGRID

Following (Campero et al., 2020), we use the same hyperparameters for all the baselines. For ICM, RND, IMPALA, RIDE and BeBold, we use the learning rate $10^{-4}$, batch size 32, unroll

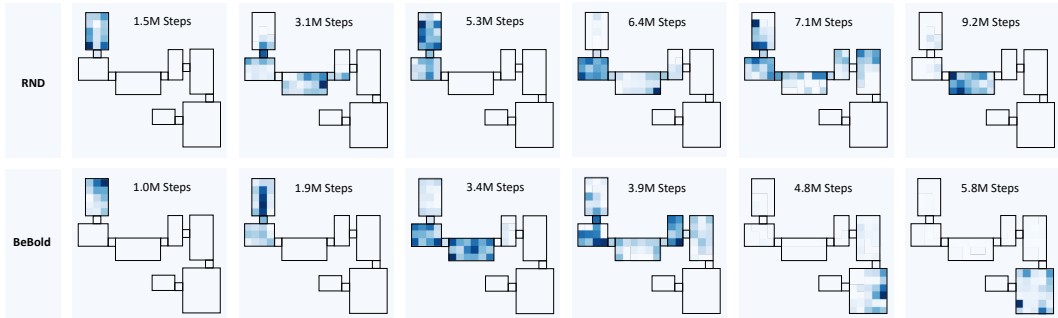

Figure 9: On policy state density heatmaps $\rho_\pi(\mathbf{s}_t)$. BeBold continuously pushes the frontier of exploration from Room1 to Room7.

length 100, RMSProp optimizer with $\epsilon = 0.01$ and momentum 0. We also sweep the hyperparameters for BeBold: entropy coefficient $\in \{0.0001, 0.0005, 0.001\}$ and intrinsic reward coefficient $\in \{0.01, 0.05, 0.1\}$. We list the best hyperparameters for each method below.

**BeBold**. For all the Obstructed Maze series environments, we use the entropy coefficient of 0.0005 and the intrinsic reward coefficient of 0.05. For all the other environments, we use the entropy coefficient of 0.0005 and the intrinsic reward coefficient of 0.1. For the hash table used in ERIR, we take the raw inputs and directly use that as the key for visitation counts.

**AMIGo**. As mentioned in (Campero et al., 2020), we use batch size of $8$ for student agent and batch size of $150$ for teacher agent. For learning rate, we use learning rate of $0.001$ for student agent and learning rate of $0.001$ for teacher agent. We use an unroll length of $100$, entropy cost of $0.0005$ for student agent and entropy cost of $0.01$ for teacher agent. Lastly we use $\alpha = 0.7$ and $\beta = 0.3$ for defining IRs in AMIGo.

**RIDE**. Following (Raileanu and Rocktäschel, 2020), we use entropy coefficient of $0.0005$ and intrinsic reward coefficient of $0.1$ for key corridor series of environments. For all other environments, we use entropy coefficient of $0.001$ and intrinsic reward coefficient of $0.5$.

**RND**. Following (Campero et al., 2020), we use entropy coefficient of $0.0005$ and intrinsic reward coefficient of $0.1$ for all the environments.

**ICM**. Following (Campero et al., 2020), we use entropy coefficient of $0.0005$ and intrinsic reward coefficient of $0.1$ for all the environments.

**IMPALA**. We use the hyperparameters introduced in first paragraph of this section for the baseline.

## C  ANALYSIS FOR MINIGRID

In addition to the analysis provided before, we also conduct some other analysis of BeBold in MiniGrid.

**On Policy State Density in MiniGrid** We also plot the on policy state density $\rho_\pi(\mathbf{s})$ for different checkpoint of BeBold. We ran the policy for 2K steps and plot the BeBold IR based on the consecutive states in the trajectory. In Fig. 9, we can clearly see that the boundary of explored region is moving forward from Room1 to Room7. It is also worth noting that although the policy focuses on exploring one room (one major direction to the boundary.) at a time, it also put a reasonable amount of effort visiting the previous room (other directions of to the boundary). Thus, BeBold greatly alleviate the short-sighted problem aforementioned.

## D  RESULTS FOR ALL STATIC ENVIRONMENTS IN MINIGRID

In addition to the results shown above, we also test BeBold on all the static producedurally-generated environments in MiniGrid. There are other categories of static environment. Results for BeBold and all other baselines are shown in Fig. 10 and Fig. 11.

*Empty* (**E**) These are the simple ones in MiniGrid. The agent needs to search in the room and find the goal position. The initial position of the agent and goal can be random.

*Four Rooms* (**FR**) In the environment, the agent need to navigate in a maze composed of four rooms. The position of the agent and goal is randomized.

*Door Key* (**DK**) The agent needs to pick up the key, open the door and get to the goal. The reward is sparse in such environment.

*Red and Blue Doors* (**RBD**) In this environment, the agent is randomly placed in a room. There are one red and one blue door facing opposite directions and the agent has to open the red door then the blue door in order. The agent cannot see the door behind him so it needs to remember whether or not he has previously opened the other door in order to reliably succeed at completing the task.

*Lava Gap* (**LG**) The agent has to reach the goal (green square) at the corner of the room. It must pass through a narrow gap in a vertical strip of deadly lava. Touching the lava terminate the episode with a zero reward.

*Lava Crossing* (**LC**) The agent has to reach the goal (green square) at the corner of the room. It must pass through some narrow gap in a vertical/horizontal strip of deadly lava. Touching the lava terminate the episode with a zero reward.

*Simple CrOssing* (**SC**) The agent has to reach the goal (green square) on the other corner of the room, there are several walls placed in the environment.

## E   HYPERPARAMETER FOR NETHACK

For general hyperparameters, we use optimizer RMSProp with a learning rate of $0.0002$. No momentum is used and we use $\epsilon = 0.000001$. The entropy cost is set to $0.0001$. For RND and BeBold, we scale the the forward distillation loss by a factor of $0.01$ to slow down training. We adopt the intrinsic reward coefficient of $100$. For the hash table used in ERIR, we take several related information (e.g., the position of the agent and the level the agent is in) provided by (Küttler et al., 2020) and use that as the key for visitation counts.

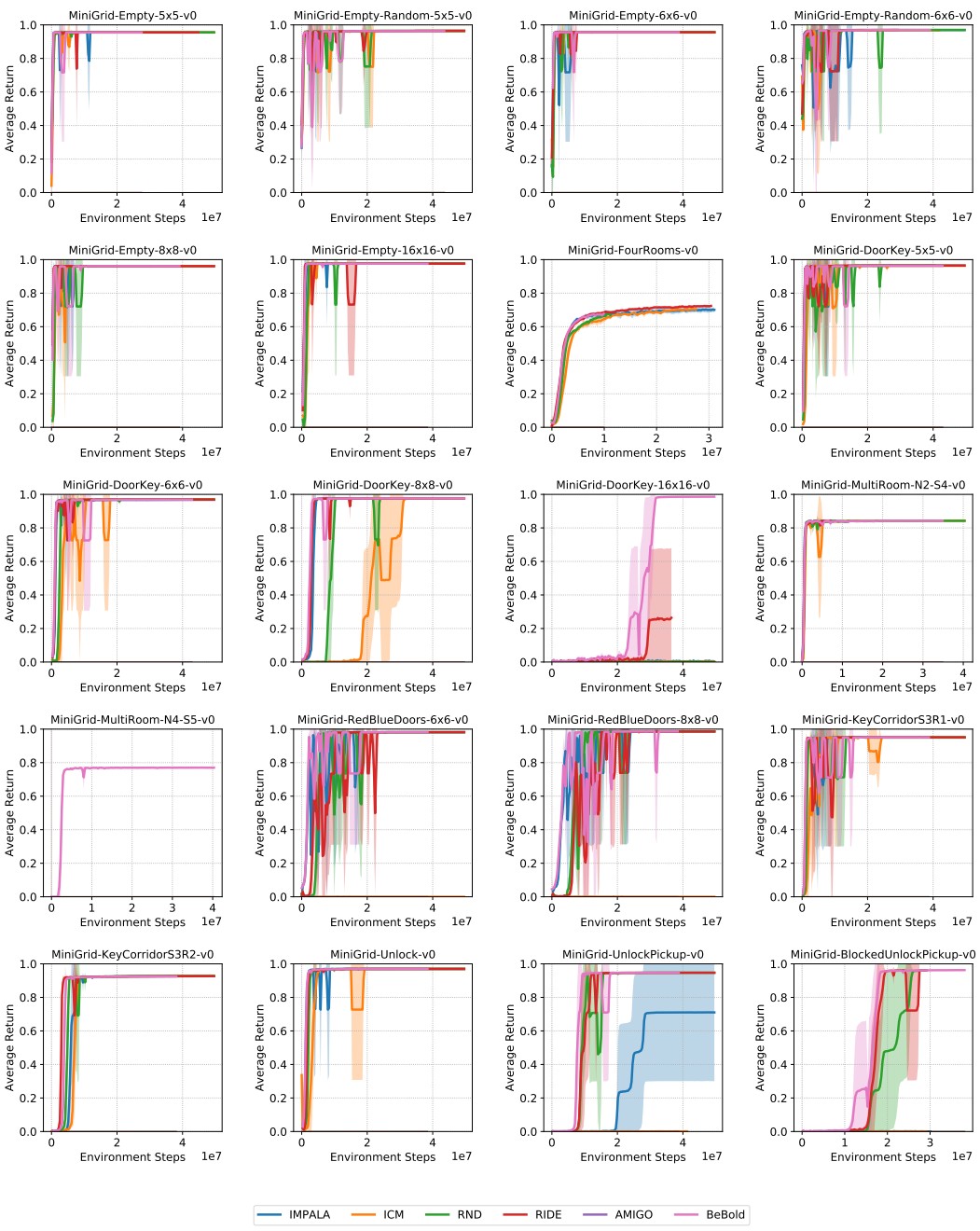

Figure 10: Results for BeBold Part 1 and all baselines on all static tasks.

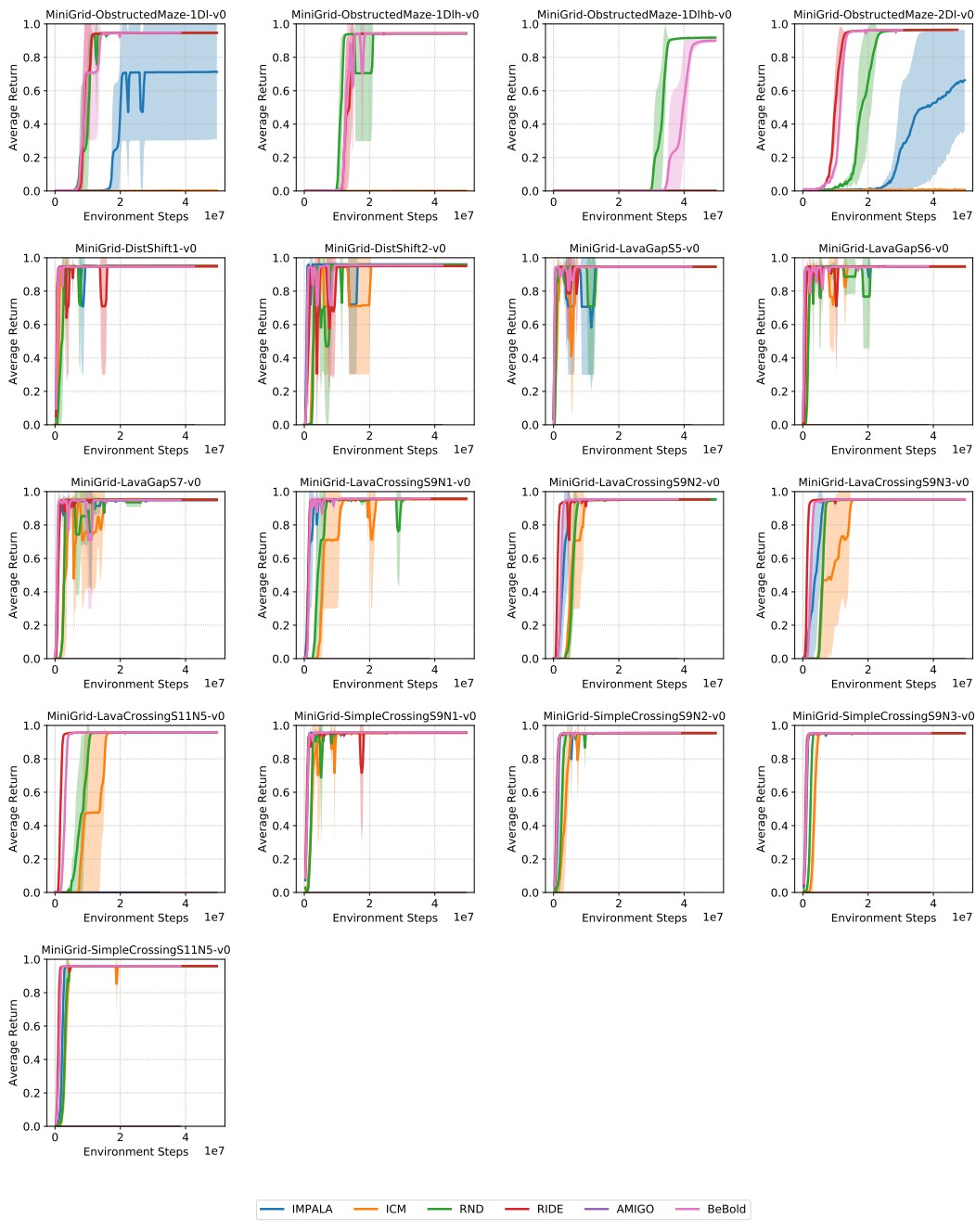

Figure 11: Results for BeBold Part 2 and all baselines on all static tasks.

