# OpenReview forum: "BeBold: Exploration Beyond the Boundary of Explored Regions"
_ICLR.cc/2021/Conference — Reject_

### Official Review · AnonReviewer4 · 2020-10-15
**This paper is overall well-written, motivated and empirically-supported.**

**Rating:** 8
**Confidence:** 5

**Review:**

------------------------------------
**Summary:**

This paper proposes BeBold, a new definition of intrinsic reward to guide exploration in sparse reward problems. This intrinsic reward combines the ideas behind count-based approaches and state-diff approaches. They demonstrate the success of BeBold by comparing their algorithm to a set of state-of-the-art exploration methods using intrinsic rewards on a set of tasks from the MiniGrid and NetHack environments.

------------------------------------
**Strong points:**

Overall, I believe this is a very good paper. The idea is simple, the motivations and intuitions are well explained. The empirical study is carefully designed, involves relevant baselines and environments. The paper is also well written. I will list the main strong points:
* The problem of exploration in sparse environments is an important problem in the field of RL. Many exploration problems have been defined (detachment, short-sightedness, etc.). This paper presents these problems and proposes to improve on some of them.
* The paper is clearly written and well organized.
* The related work is relevant and seems complete.
* The experiments are overall well designed. MiniGrid and Nethack are two types of benchmarks adapted to the study of exploration algorithms and the baselines are relevant. I like that the authors conducted ablations studies and additional experiments to study specific aspects of their strategy (visitation counts analysis and the study of the short-sighted problem).
* The new methods seem to significantly outperform previous ones without being overly complicated. The empirical evidence supports the claim (although I have a problem with the use of “solve” here, see below).

------------------------------------
**Weak points:**

I will now list a few weak points of the paper.
* The paper lists various exploration problems (noisy TV problem, detachment, derailment, short-sightedness and asymptotically inconsistent). BeBold is said to solve detachment, short-sightedness and asymptotical inconsistency. Derailment and the noisy TV problem are neither defined nor discussed. I believe BeBold does not solve the noisy TV problem, it should be discussed.
* I am not sure I get the intuition behind the clipping of the reward function. The authors write: “we do not want to give a negative IR to the agent if it transits back from a novel state to a familiar state”, but do not really explain why. Can you make this explicit?
* Episodic restriction seems to restrict the use of BeBold to environments with discrete countable states (use of hash table). Do you see a way around that problem?
* I think the main problem is the total absence of discussion w.r.t. the potential limits of BeBold (there is no discussion section). In particular, the authors could discuss the aspects mentioned above: the limitation to discrete states, the noisy TV and derailment problems.
* Comparison to Go-Explore: Talking about BeBold, the paper says: “without using domain knowledge (e.g., image downsampling like in Go-Explore (Ecoffet et al., 2019)). It is true that BeBold does not use image downsampling, but it does not use images at all, which is probably the explanation. Go-Explore and RND can be applied to image-based environments like Montezuma’s revenge. To do so, Go-Explore indeed requires downsampling. However, I believe that if BeBold were to be applied to such environments, it would also need the downsampling trick (or equivalent) to be able to apply episodic restriction.
* 4 seeds is a very small number. I know that very few works present much more than this, but I think it is bad practice. Please consider adding more (>=10). I admit that in your case, the trend is quite clear as competing algorithms show null performance most of the time.
* I think the repeated use of the term “solved” is quite misleading. The traditional definition of “to solve” is probably “to reach maximum performance”. This is definitely not the case in several environments. Please consider either defining precisely what you mean by “solved”, or not using that term.
* There is no codebase in the supplementary material. Do you plan on releasing it? Please consider doing so.

I believe most of these weak points can be solved during the rebuttal.

------------------------------------
**Recommendation and justification:**

I think this paper should be accepted for the reasons listed in the Strong Points section. I'm giving a score of 7. I'd be happy to raise this score to an 8 provided that the authors add a discussion section to discuss the points mentioned above.

------------------------------------
**Feedback to improve the paper (not part of assessment):**

* “in which the agent gets trapped into one (long) corridor and fails to try other choices.” →  not very clear at this point, although it’s explained later.
* The episodic restriction seems to require a policy that has memory. Otherwise the reward would not be Markovian anymore. Maybe this should be said somewhere?
* Short-sightedness paragraph: “but it is often far less than enough in a complex environment”: vague and not argumented.
* What do the environment code stands for? S, R, Dl-h, Dl-hb, Q?
* Results could be a little bit bigger, maybe switch the colors so that the two leading algorithms do not have almost identical colors (pink and red)?
* In tables, what does bold indicate? Is it only the best value? does it assert statistical significance (if yes, which test at which confidence level?)
* I believe the paper does not mention which RL algorithm is used. This seems like an important detail to provide.

**Typos:**
* “In contrast, (Campero et al. 2020) solves 50% of the tasks…” →  remove parentheses. Same in the last paragraph before Sec. 5 “For this, (Zhang et al, 2019) ...  (Raileanu)...”. Other examples throughout.
* Reward definition in Sec2. Not well defined, we don’t know what are t, k or what the =1 refers to.
* Do not use contractions “does’nt”, “don’t”, “won’t”, “let’s” etc.
* Sec. 3, Episodic Restric. paragraph: “back and force” → “back and forth”
* “the observation to the agent” → “the observation of the agent” ?
* Table 2 caption: “th” → “the”.
* Second to the last paragraph before the Intrinsic Reward Heatmap paragraph: “discover” → “discovers”.
* “BeBold doesn’t suffer from short-sighted problem” → “BeBold does not suffer from short-sightedness” or “the short-sighted problem”.
* “We only give RND intrinsics” → is “intrinsics” a word ?


**Post-rebuttal update**
The authors addressed most of my concerns during the rebuttal, added relevant discussions, experiments on Montezuma and clarified the reported evaluation metrics. I raise the score from 7 to 8.

---

> ### Author Response · Authors · 2020-11-12
> **Author Response for R4**
>
> We thank R4 for all the comments. Please also refer to the common questions above for the answer to the remaining questions
>
> **Q1**: The number of seeds (4) used for experiments is very small.
>
> **A**: We thank the reviewer and will update the results accordingly if the paper gets accepted. We also note that around 4 seeds is indeed quite common in measuring uncertainty. Please see that RND[1] uses 5 seeds, RIDE[2] uses 5 seeds, AMIGO[3] uses 5 seeds, etc.
>
> **Q2**: The usage of the term “solved” in MiniGrid environments is quite misleading.
>
> **A**: We follow the default environment specifications in MiniGrid. In the MiniGrid environments, the agent only receives a reward of (1-#steps_taken/max_steps) when it reaches the goal and then the episode ends (otherwise the agent receives 0 rewards). This means that if the reward is positive, then the agent has reached the goal and the task is considered "solved". Please note that the maximum reward is not 1 for MiniGrid.
>
> In addition to that, in Fig. 3 and Fig. 6, we are reporting training reward averaged over 4 runs and 100 episodes. In the figure, the action is sampled from the distribution output by the policy network instead of taking max over that. Such randomness might also affect the final performance of the agent.
>
> **Q3**: Episodic restriction seems to restrict the use of BeBold to environments with discrete countable states.
>
> **A**: We think that this is a valid concern. For continuous states, we could discretize the space and still use a hash table to count state visitations. We did that for MonteZuma’s Revenge. We leave the more general solution to future work.
>
> **Q4**: Do you use any domain knowledge as Go-Explore did (e.g., in MonteZuma's Revenge)?
>
> **A**: Two kinds of domain knowledge exist in Go-Explore: 1. They downsample the image and put them into cells, and assume states that the agent is in the same cell are equivalent. 2. They use the representation (room, x coordinate, y coordinate) to represent the goal and calculate the novelty.
>
> Note that for our experiments on MonteZuma’s Revenge, we don’t use any domain knowledge as in Go-Explore. The episodic restriction also doesn’t use any downscaling, just raw pixels.
>
> **Q5**: Do you plan to release the code?
>
> **A**: We’ll definitely release the code afterwards.
>
> **References**
>
> [1] Burda, Yuri, et al. "Exploration by random network distillation." arXiv preprint arXiv:1810.12894 (2018).
>
> [2] Raileanu, Roberta, and Tim Rocktäschel. "RIDE: Rewarding Impact-Driven Exploration for Procedurally-Generated Environments." arXiv preprint arXiv:2002.12292 (2020).
>
> [3] Campero, Andres, et al. "Learning with amigo: Adversarially motivated intrinsic goals." arXiv preprint arXiv:2006.12122 (2020).
>
> [4] Ecoffet, Adrien, et al. "Go-explore: a new approach for hard-exploration problems." arXiv preprint arXiv:1901.10995 (2019).
>
> [5] Ecoffet, Adrien, et al. "First return then explore." arXiv preprint arXiv:2004.12919 (2020).
>
> [6] Guo, Yijie, et al. "Self-Imitation Learning via Trajectory-Conditioned Policy for Hard-Exploration Tasks." arXiv (2019): arXiv-1907.

---

> > ### Comment · AnonReviewer4 · 2020-11-13
> > **Response**
> >
> > I appreciate this structured and detailed answer.
> >
> > **Q1**
> > I will trust you on this. As I said, I know most approaches use such small amounts of seeds. I believe this is not an argument. An argument would be to demonstrate that this can be enough to prove the type of effect size you expect. A posteriori, this might work because your effect sizes are large. 5 seeds, however, is most of the time far from sufficient, see [1], [2].
> >
> > **Q2**
> > This sounds weird to me. The standard practice is to evaluate agents without exploration noise. Can you justify why you did something different here? You are saying that the term 'solve' refers to results that are not presented in the paper, how should we know? If you think evaluation should include exploration noise, then certainly the paper should describe these results. The easiest way to solve the confusion, I believe, is to plot the testing results without exploration noise. This is standard practice, and would illustrate the term "solve" that is used throughout the paper.
> >
> > **Q3 & Q4**
> > I thought you would have to downsample the state to actually get a meaningful hash function. Since you do not, my comment is not relevant anymore. Thanks for the precision.
> >
> > **Q5**
> > This is great.
> >
> > About the main answer above, it answers some of my points, but it is not clear whether these points will be explicitly discussed in the paper. I would like to see a discussion section presenting the limits of the approach (noisy tv, derailement, , limitation to discrete states). The integration of the preliminary results on Montezuma would also be a plus, as it demonstrates that BeBold can handle pixel-based inputs.
> >
> > I believe I only need further discussion of Q2. Thanks again for the discussion.
> >
> > [1] Henderson, P., Islam, R., Bachman, P., Pineau, J., Precup, D., & Meger, D. (2017). Deep reinforcement learning that matters. arXiv preprint arXiv:1709.06560.
> > [2] Colas, C., Sigaud, O., & Oudeyer, P. Y. (2019). A Hitchhiker's Guide to Statistical Comparisons of Reinforcement Learning Algorithms. arXiv preprint arXiv:1904.06979.

---

> > > ### Author Response · Authors · 2020-11-13
> > > **Author Response for R4**
> > >
> > > We thank R4 for the comments.
> > >
> > > **Update on the paper**: We are currently working on the training of MonteZuma’s Revenge now. The discussion on the limitations and the updated results is on the plan when we finish MonteZuma’s experiments.
> > >
> > > **Q2**: Thanks for the suggestion and we think that this is indeed a valid concern. We report the training performance following RIDE [2] and AMIGo [3]. We can also update the paper with a table of final testing performance of each method without exploration randomness in the next revision. We think that although training performance is less accurate, it could be a good indication of the testing performance in MiniGrid (especially on many maps, the baselines all get zero training reward but BeBold achieves pretty good performance).

---

> > > > ### Comment · AnonReviewer4 · 2020-11-16
> > > > **R4 response**
> > > >
> > > > **Q2**: Ok, now I understand that you plot the training performance. Although training performance might be a good indication of the testing performance, the direct testing performance is definitely its best own proxy. I believe providing training performance is not a very good practice. First, it does not evaluate the true performance of the agent in testing conditions. The effect of the exploration noise is unclear. Second, it is a running average of the $N$ past training performance points. This means that a given point does not evaluate a policy but $N$ policies. Evaluating the agent offline by running $M$ evaluation episodes is usually very cheap computationally compared to the cost of training, why not do it and show on the graph what we want to see there, instead of a proxy?
> > > >
> > > > Given that all experiments are already run, adding a table with final testing performance (central tendencies and errors measures) will be sufficient. However, please consider running proper evaluations in the future.

---

> > > > > ### Author Response · Authors · 2020-11-18
> > > > > **Author Response for R4**
> > > > >
> > > > > **Q2**: We thank the reviewer for the comments. We are currently working on providing a table with the final performance. We are also planning to rerun experiments with proper evaluations if the paper gets accepted.
> > > > >
> > > > > Please also note that we made a small mistake in the previous reply on the reward definition of MiniGrid: In the default MiniGrid specification, the agent receives the reward of (1 - #steps_taken/max_steps) if it reaches the goal (otherwise 0). So the maximum reward is not 1 and we consider a positive reward as "solve" the environment. This also explains why on some tasks in Fig. 3 and Fig. 6, the reward is not 1.

---

### Official Review · AnonReviewer2 · 2020-10-27
**This paper offers a relevant contribution to address the problem of properly exploring the known state-space by providing an intrinsic reward for transitioning from highly visited to poorly-visited states.**

**Rating:** 9
**Confidence:** 4

**Review:**

BeBold: Exploration Beyond the boundaries of Explored Regions

The authors address the problem of exploring efficiently for a reinforcement learning agent when the reward function is sparse. They propose to compute an intrinsic reward based on the inverse visitation count to reduce the visit imbalance generated by optimizing for the extrinsinc reward.

This work present an interesting approach to progressively explore at the boundaries of the most visited states (a phenomenon that occurs when the agent focus on maximizing known rewards). They address the key problem of short-sightedness in this domain. The contribution is clearly stated by the authors. The results support the claim that their method achieved state-of-the-art performance of the Minigrid environments and Nethack. They also provide an analysis of the necessity of clipping and having an episodic restriction on the intrinsic reward (ERIR).

A couple of remarks however:
- the context should be clarified: you mention a quite classical RL setting, which assumes is done by interacting with the environment. In that case, one can explore via action selection and potentially discover new states (by trying new actions or experiencing an unexpected outcome). On the other side, you method tackle the problem of visiting properly a known space (as the effect of the method "marks" actions that transition from well-visited to poorly visited parts of the state space for the next episode). This should be made explicit.
- How is this augmented reward information used? I didn't see a clear mention of the learning algorithm implemented in the agent, please make that explicit.

As a main criticism, I's day that the underlying assumption of this work is that the environment is somewhat known and that "exploration" only happens within this known environement. It is critical to the method that the action selection process itself moves the agent towards fully-unknown parts of the state space.

Section 6 should be clarified on how your methods and the related work actually relate ; it is only done for the last paragraph but should be clarified for the other approaches you cite.

Typos:
p4: "from MiniGird: ..."
p6: "... heatmap of normalizd visitation counts..."
p8: "... maximizng mutual information"

---

> ### Author Response · Authors · 2020-11-12
> **Author Response for R2**
>
> We thank R2 for all the comments. Please also refer to the common questions above for the answer to the remaining questions
>
> **Q1**: The environment is somewhat known and that "exploration" only happens within this known environment. It is critical to the method that the action selection process itself moves the agent towards fully-unknown parts of the state space.
>
> **A**: We emphasize that the agent only receives partial observation and the environment is indeed unknown to the agent. We admit that the algorithm also to some extent relies on random actions to discover new states like most other RL algorithms (PPO, DQN, ICM, RND, etc). But the key difference is that once the agent randomly steps out of the boundary, discovering a new state, it will receive a high IR and thus will be encouraged to reach here again. Such “structured” exploration (encouraging the agent to explore beyond the boundary) cannot be done using random action. The advantage of BeBold over random action selection exploration is clearly shown in the experiments as IMPALA fails in most of the MiniGrid environments.
>
> **Q2**: How is this augmented reward information used?
>
> **A**: The augmented reward (intrinsic reward, IR) will be received by the agent along with the extrinsic reward (ER). The policy of the agent will be trained using PPO to reach maximum combined reward (IR + ER).

---

### Official Review · AnonReviewer1 · 2020-10-27
**Well motivated, strong empirical results**

**Rating:** 7
**Confidence:** 4

**Review:**

Summary

The authors propose a novel intrinsic reward based on the difference of inverse visitation counts for consecutive states. This reward encourages the agent to explore beyond the boundary of already explored regions. Using a few simple examples, they show that the proposed intrinsic reward mitigates the problems of detachment and short-sightedness which are common for count-based methods. The method shows superior performance on a number of tasks from two procedurally-generated benchmarks, MiniGrid and NetHack. The paper also contains comparisons with a few strong baselines (including SOTA on these benchmarks), analysis of the learned behavior and intrinsic reward, as well as ablations of the proposed approach.


Strengths

Overall, I really liked this paper. The proposed method is simple, well motivated, clearly explained, significantly outperforms the SOTA, and solves hard exploration tasks that were previously out of reach. I found the empirical evaluation to be very thorough and comprehensive, including multiple baselines and ablations. I particularly liked the careful motivation of this approach together with the quantitative analysis in Section 5.2 (which compares the visitation counts and intrinsic rewards of different methods). In addition, the authors support their claims regarding the short-sightedness and detachment issues with well designed experiments and metrics.


Weaknesses

One thing that wasn’t clear to me after reading the paper was what are the teacher and student networks used to approximate the visitation counts. Are these equivalent to the predictor and random network used by RND? This is an important detail so please clarify.

Could the network phi (used to estimate visitation counts) “forget” previously visited states and thus lead to short-sighted behavior in a similar way as count-based methods do (i.e. oscillate between two state regions / corridors)? Have you ever observed this behavior in practice and if not, do you have any intuition why?

Have you tried scaling the IR reward by the inverse of the episodic state counts (like RIDE does) instead of only using the episodic restriction? It might be interesting to add it as an ablation and give some intuition on which (when) one is preferable.

I think it would be valuable to include all the baselines in the analysis section i.e. Tables 1, 2, and Figures 4, 5 (they can be in the appendix if there isn’t enough space).

Can you specify how many seeds you used for computing the mean and std in the plots? I could not find information and it is important in order to understand the significance of the results.


Minor Points

Is there any reason for which in Figure 4 you don’t show the visitation counts for both models for the same number of environment steps? I think that would be a more clear way of presenting those results.

There is some inconsistency in the notation used. In Figure 1, you use c(s) to denote visitation counts, while in Section 3, you use N(s).

Typos:
Table 2: “entropy of the”
Figure 2: “MRN6” → MRN6-SX?
Citations in various places contain parentheses where they should not (i.e. in the middle of a sentence).


Recommendation

This paper presents a novel and effective method for an important problem, and it also provides insightful analysis to better understand the limitations of different approaches. Thus, I think it would be a valuable contribution and I recommend it for acceptance.

---

> ### Author Response · Authors · 2020-11-12
> **Author Response for R1**
>
> We thank R1 for all the comments. Please also refer to the common questions above for the answer to the remaining questions
>
> **Q1**: Could the network $\phi$ (used to estimate visitation counts) “forget” previously visited states and thus lead to short-sighted behavior in a similar way as count-based methods do (i.e. oscillate between two state regions/corridors)?
>
> **A**: We haven’t seen this issue in our experiments. Our hypothesis is that maybe the network capacity is enough.
>
> **Q2**: Have you tried scaling the IR reward by the inverse of the episodic state counts (like RIDE does) instead of only using the episodic restriction?
>
> **A**: Thanks for the suggestion. That’s a good point. We will conduct that ablation study and update the paper.
>
> **Q3**: It would be valuable to include all the baselines in the analysis.
>
> **A**: Thanks for the suggestion. We will try to include that in our paper.
>
> **Q4**: Can you specify how many seeds you used for computing the mean and std in the plots?
>
> **A**: We use 4 seeds to compute the mean and std in the plots.
>
> **Q5**: Is there any reason for which in Figure 4 you don’t show the visitation counts for both models for the same number of environment steps?
>
> **A**: Because some of the checkpoints are similar to the previous ones (indicating that the agent is exploring the same room) and not representative enough to illustrate the problem, we choose to present the checkpoint at different time steps.
>
> **Q6**: There is some inconsistency in the notation used ($c$ and $N$).
>
> We thank the reviewer for pointing out. We’ll update the paper
>
> **Q7**: Are teacher/student networks equivalent to the predictor and random network used by RND?
>
> **A**: Yes we follow the paradigm in RND: the teacher network is equivalent to the random network and the student network is the predictor network.

---

> > ### Comment · AnonReviewer1 · 2020-11-15
> > **Reviewer Response**
> >
> > Thank you for the clarifications. I look forward to seeing the updated draft with the added ablation and analysis.

---

### Official Review · AnonReviewer3 · 2020-10-29
**Updated review for Submission 2124**

**Rating:** 4
**Confidence:** 4

**Review:**

**Summary**
This paper is a presentation of BeBold, a new method using an intrinsic reward for exploration, meant for procedurally generated, episodic environments. The method includes two major components: the first being intrinsically rewarding the agent for entering states that are less visited than the current state and the second being only intrinsically rewarding the agent for the first time it enters a state during an episode. The agent's intrinsic reward is larger if the difference in visit counts is larger. The paper includes some discussion of the conceptual advantages of BeBold over prior work as well as empirical demonstrations in MiniGrid and NetHack.

**Strengths and Weaknesses**
The primary strength of this paper is the BeBold reward as an idea. The most noteworthy weaknesses, from my point of view, are the discussion of "short-sightedness" and the narrative used to describe BeBold, both frequently seeming misleading or missing important clarifying points. However, the paper is supported by other strengths including an appropriate discussion of related work.

The authors have made short-sightedness a core support for their proposed method. Unfortunately, they have not provided sufficient evidence to convince me that short-sightedness is necessarily a problem. The intuition with the corridor example makes sense, and some of the experiments seem to support the idea. However, it’s not clear to what extent this is a serious issue and how much it affects exploration. Furthermore, the other experiments seem to contradict the authors’ reasoning and cast doubt on their analysis. The pseudo-mathematical analysis seems seriously oversimplified and flawed, since it makes several incorrect assumptions and still doesn’t prove the authors’ point more than the intuitive explanation. The experiments also have issues that prevent them from providing appropriate support.

**Recommendation**
I am recommending that the paper be rejected primarily because, as stated, insufficient evidence is provided for the core support for the paper.

**Specific Examples of Issues**

1. Conceptual Issues:

1.1 It is implied that the second-longest corridor will receive the second-largest amount of exploration, but this is not true in the results shown in Table 1, where the second-longest corridor (length 30) is never the corridor with the second-largest amount of exploration. This disagreement is concerning and brings doubt into the entire discussion.

1.2 It is not generally true that "the probability to take action left or right is proportional to the accumulated reward" (p. 12) This assumption is not the case for epsilon-greedy Q-learning nor softmax/Boltzmann exploration (at least for most temperature settings), which are two very common strategies.

1.3 The description of short-sightedness in the introduction ("the agent often settles in local minima, sometimes oscillating between two states that alternately feature lower visitation counts" (p. 1) does not clearly match the problem of short-sightedness described in Section 4, where it appears to refer to the problem where longer trajectories (corridors) provide more accumulated intrinsic reward.

1.4 The method is designed for procedurally generated environments, and there is no discussion of and no experimental work dealing with stochastic environments. While it is completely acceptable to design a method that works for one type of environment and not another, the reason I see this omission as slightly problematic is that all of the primary comparisons are with count-based intrinsic rewards. One of the explicit strengths of count-based rewards is that they encourage the agent to return to states multiple times to acquire enough data to account for high variance. Do you know if it is possible that such environments might give count-based methods the edge over BeBold? While testing this hypothesis is beyond the scope of this paper, I think a comment about such a possibility would be appropriate to include in this paper.

1.5 It appears that the example 1/N count-based bonus that is used (Section 3 and Appendix A) to demonstrate a conceptual disadvantage of count-based methods over BeBold is not used in the works you cite. Bellemare et al (2016), Ostrovski et al. (2017), and Puigdomènech Badia et al. (2020) all use 1/$\sqrt{N)}$-esque bonuses and the exact decay rate of RND (Burda et al., 2018b) is difficult to put one's finger on, but it isn't designed to mimic 1/N. In this way, using a 1/N bonus seems inappropriate for demonstrating a conceptual shortcoming in existing work. This is an important concern for me, because it comes across as misinforming the reader.
Note: I believe the citation to Burda et al. (2018a) should be with the citation to Pathak et al. (2017), rather than with the count-based methods.

2. Weaknesses in the experiments:

2.1 Why use a Q-network for the tabular setting at all? To provide an empirical demonstration of the discussion in Appendix A, initializing all Q-values to zero and updating them as discussed in the Appendix would likely be more instructive for the reader.

2.2 Because there is randomness involved (for example, in the Q-network as mentioned on page 6) I expect there to be variance in the numbers reported, but Tables 1 and 2 do not appear to include any measure of uncertainty. How many runs were done? Further, I can't find an explanation of how the Q-network was initialized, including what kind of random initialization was used.

2.3 Did BeBold solve all of the environments or only the 12 "most challenging ones"? "We test BeBold on all environments from MiniGrid. BeBold manages to solve the 12 most challenging environments" (p. 5). I found the way this was expressed a touch unclear. I'm hoping that you could instead write, "BeBold manages to solve all environments, including the 12 most challenging environments."

2.4 In Table 2 (p. 6) there is no mention of what 0.2M, 0.5M, etc. mean. Are they episodes?

2.5 While the introduction asserts that there are quantitative results showing that BeBold mitigates the detachment problem (p. 2), I could not find any discussion of detachment in the discussion of the quantitative results in the paper. I'm also a bit doubtful that the included experiments are appropriate for demonstrating this mitigation quantitatively.

2.6 Can you provide a little more explanation about why clipping needs to be done to allow for a fair comparison? Does clipping mean the same thing in this context as it did in Section 3, that is, to only allow positive rewards? "We remove clipping from BeBold for a fair comparison" (p. 6).

2.7 I'm concerned about the choice to measure the quantity of training data based on the number of episodes rather than the number of time steps. As Machado et al. (2018, Section 3.1.3, pp. 528-529) suggest in their discussion of benchmarking reinforcement learning agents, when episode lengths differ from episode to episode, the policy affects the amount of training data. Do you have a conceptual reason to think that this won't pose a problem in your experiments?
Machado, M. C., Bellemare, M. G., Talvitie, E., Veness, J., Hausknecht, M., & Bowling, M. (2018). Revisiting the arcade learning environment: Evaluation protocols and open problems for general agents. Journal of Artificial Intelligence Research, 61, 523-562.

3. Inaccurate references to the literature:

3.1 It is not obvious how the overall goal of Brockman et al. (2016), which I read to be a presentation of OpenAI Gym, supports the point you are making: "most work requires either a manually-designed dense reward" (p.1) I can see it supporting the point that much work makes use of manually-designed dense rewards, but not that not having it causes most algorithms to fail. It may be helpful to provide a page number so the interested reader can more easily follow your thinking. I was expecting some kind of survey or argument about how the type of reward affects performance in Deep RL, so this citation was surprising for me.

3.2 Detachment and derailment are written as though they are a single concept (p. 1) and the example doesn't really give a very accurate sense of the definitions of either. What does this example have to do with derailment?

3.3 The explanation of detachment on page 4 does not seem to accurately capture the meaning as it was defined by Ecoffet et al. (2019). In particular, an important property of a detachment situation is that the intrinsic reward in an area between two unexplored areas is exhausted, meaning that the exploration of one of those areas is not only delayed, as in your example, but is never induced through intrinsic reward because there isn't enough intrinsic reward left to shape behaviour towards that area.

4. Incorrect or missing logic:

4.1 The paper claims that another method relies on good-quality generalization because it uses two stages. This is not necessarily true.

4.2 How can you be sure that reliance on quality generalization is the reason that BeBold performs better than RIDE? ("As a result, BeBold shows …")

4.3 Why are you comparing your method with Campero et al. (2020) rather than any of the other examples that use MiniGrid? "In contrast, (Campero et al., 2020) solves 50% of the tasks, which were categorized as “easy” and “medium”, by training a separate goal-generating teacher network in 500M steps" (p. 2) I suspect it is the present state-of-the-art at the time of writing, but would appreciate the narrative in the paper confirming that.

5. Failure to communicate the connection between other sources and this paper:

The paragraphs on Go-Explore, Frontier-based exploration, and BeBold comment separately on each approach without relating them. This part is especially difficult to understand since the reader hasn't been told very much about how BeBold works at this point in the paper.

For example, it may be useful to clarify that because the Frontier-based methods described are designed for robots, they take advantage of properties of the physical spaces that those robots are designed for, like being structured such that exploring a 2D map is a reasonable goal.

6. References to ideas and existing literature that haven't been introduced:

6.1 At the point in the text where the reader is pointed to Figure 1, (the end of page 1) RND has been introduced as a module of the proposed method used to approximate the inverse visitation count, not as a competing method, and no explanation of how RND is designed has been provided, with the same going for any clues as to why RND gets "trapped." The function c is not defined.

6.2 The term trajectory should be defined for your context. In some papers, a trajectory is any sequence of state-action-rewards in the environments, while in others it only includes those corresponding with a complete episode. It would probably be best to try to include a definition around its first use on page 1: "Finally, state-diff approaches offer rewards if, for each trajectory, representations of consecutive states differ significantly." Some kind of explanation of why trajectories get so much focus in the early part of the paper would also be helpful.

**Additional Feedback (Here to help, not necessarily part of decision assessment)**

One way to make this paper easier to follow, and perhaps more impactful on the reader, would be to order the paper differently. Because the algorithm is so much inspired by the gaps in the literature, I think explaining the shortcomings of different existing algorithms before explaining BeBold might work really well. You could leave it in the current order, but the story needs to be told quite differently to avoid the many references to existing literature without explanation.

The phrase "criterion for IR" (in the Abstract) made me think it meant some kind of requirement for something to be defined as IR, and from context, I think you're not trying to define IR, you're trying to define a new type of IR, so I would consider rephrasing. You used "exploration criterion" later and it is a bit less confusing. Matching the wording of other papers who are creating new types of IR would improve understanding within the group of researchers focused on IR.

"The statement "Random exploration (e.g., $\epsilon$-greedy) in these environments is often insufficient and leads to poor performance (Bellemare et al., 2016)" (p. 1) leaves room for misunderstanding from the reader, so I recommend something like "Random exploration (e.g., $\epsilon$-greedy) __alone__ is often insufficient."

"have proposed to use intrinsic rewards" (p. 1) → "have proposed using intrinsic rewards"?

"This addresses asymptotic inconsistency in the state-diff, since the inverse visitation count vanishes with sufficient explorations." I expected the word "exploration" rather than "explorations" here because the word "exploration" is more commonly used as an uncountable noun (meaning the plural form is still "exploration") but if you mean exploration in a sense that is countable, a definition would be really helpful. (p. 1)

"transition function T : S × A → P(S) where P(S) is the probability of next state given the current state and action." This is kind of strange notation since P(S) doesn't seem to be a set of anything. Also, it doesn't seem like T appears anywhere else in the paper, so you probably don't need this definition at all. (p. 2)

To help clarify this reward, it would be useful to discuss when the counts, $N(s)$, are updated, which I'm guessing is at the end of the episode? The definition in (1) doesn't result in the properties described if the counts are updated online after each transition (in which case, if we imagine a corridor, then $N(s_{t+1}) = N(s_t) - 1$, so you get the same properties of "short-sightedness" as offered by count-based methods.

"We clip the IR here because we don’t want to give a negative IR to the agent if it transits back from a novel state to a familiar state." (p. 3) Can you explain briefly why this would be problematic? I think it is natural to ask why we shouldn't just keep pushing the agent forward into the frontier by giving it a negative reward if it takes a step back? An example of a situation where such a choice would cause a problem would be helpful here.

"As a result, the agent will continue pushing the frontier of exploration in a much more uniform manner than RND and won’t suffer from short-sightedness." RND hasn't actually been mentioned in the text prior to this point, so I think this would be quite confusing for someone who does not know RND very well.

"partial observation o_t are used" This looks like a typo. (p. 3)

"RIDE (Raileanu and Rocktäschel, 2020) avoids this by scaling the intrinsic reward r(s) by the inverse of the state visitation counts." I don't think you use the notation r(s) ever again.

"The corridor $\tau_j$ has a length of $T_j$." Since you never use $T_j$ again, as far as I can tell, I recommend just telling the reader that the lengths of the corridors vary to avoid adding notation that isn't used. (p. 3)

"preference on exploring" should probably be "preference for exploring" (p. 3)

"Automatic curriculum approaches (Campero et al., 2020)) have similar issues due to an ever-present IR." Extra parenthesis. (p. 4)

"For this, (Zhang et al., 2019) proposes to learn a separate scheduler to switch between intrinsic and extrinsic rewards, and (Raileanu and Rocktäschel, 2020) divides the state representation difference by the square root of visitation counts." (1) Accidental citep instead of citet, (2) should probably be "proposes learning" rather than "proposes to learn" since it is the agent that will be learning, rather than the authors, and (3) should be divide rather than divides since Raileanu and Rocktäschel are two people. (p. 4)

"We only evaluate AMIGo for 120M steps in our experiments, the algorithm obtains better results when trained for 500M steps" There's a comma splice there; it would be helpful to have some kind of transition to help the reader leap between the two ideas and why they are shared here. (p. 4)

"where each tile contains an object" Should this actually be "where each tile may contain an object"? (p. 4)

"MR consists of a series of rooms connected by doors and the agent must open the door to get to the next room." I assume they have to open more than one door in a single environment? I was expecting, "the agent must open each door to get to the subsequent room" or something along those lines. (p. 4)

In Figure 3, I think some kind of indicator of the maximum possible return would be helpful. (p. 5)

"Multi Room environments are relatively easy in MiniGrid." I don't understand what this means. Do you mean that they are relatively easy for BeBold, and if so, how are you measuring "easiness"? (p. 5)

"RND, AMIGo, RIDE, IMPALA and BeBold successfully solves KCS3R3" should probably be "RND, AMIGo, RIDE, IMPALA and BeBold successfully solve KCS3R3" (noun-verb agreement, p. 5)

"We analyze how BeBold mitigates the short-sightedness issue by only using IR to guide exploration" (p. 5) From context, I'm guessing that you mean that you are no longer including the extrinsic reward, so in this section ($r_t = r_t^i$)? The phrase "to guide exploration" is a little ambiguous, so you might do better to be more specific, like "only using IR in the computation of the value function" or something along those lines.

"Entropy of th visitation counts" (p. 6) Typo in the caption for Table 2.

"give all states equal amount" (p. 6) Missing article on equal amount (probably should by "an equal amount")

"normalizd visitation counts" (p. 6) Typo.

"discover all the rooms" (p. 6) should probably be "discovers all the rooms"

I was a bit uncomfortable with the definition of curiosity-based intrinsic reward, though it is appropriate for all of the intrinsic reward methods that use the word curiosity that I am aware of, because the term curiosity has been used quite differently for other methods that don't use an intrinsic reward, like the method published by Still and Precup (2012).
Still, S., & Precup, D. (2012). An information-theoretic approach to curiosity-driven reinforcement learning. Theory in Biosciences, 131(3), 139-148.

Your bibliography has some capitalization errors. For example, Vinyals et al. 2019 has lost the capital S, C, and II for StarCraft II, Silver et al. 2016 and 2017 have lost the capital N in Nature, and Brockman et al. 2016 has lost the capital AI in OpenAI Gym.

"Both starts with" probably should be "Both start with" (noun-verb disagreement, p. 12)
"If an agent have" probably should be "If an agent has" (noun-verb disagreement, p. 12)
"Using count-based approach" probably should be "Using a count-based approach" (missing article, p. 12)

-------------------------------------------------------------------------------
**UPDATE from responses**

Thank you for your detailed response to my comments.

The majority of my comments describing the weaknesses of this paper are related to both the conceptual and experimental components of the corridor example, so I want to recommend not including the example at all. It does not add clarity or provide any conceptual insights. This is in agreement with your stated primary focus, which is on demonstrating the empirical performance of a novel intrinsic reward design. Further, if you removed nearly all of the parts of the paper referring to the corridor example, you might have space to explain the detachment and derailment and the relationship of your method to these problems, as well as the other terminology that is currently not defined. It may be appropriate to include the idea briefly as motivation.

**It is not clear to what extent short-sightedness is a serious issue and how much it affects exploration.**

None of the papers have the term "short-sighted" in them, so at the very least, you are using a new name for a problem and then not defining it clearly. To the best of my reading, the problem you describe as short-sightedness is not described in these papers. Ecoffet et al. focus on derailment and detachment (which are different from the short-sightedness problem described in this paper) and while Guo et al. do describe myopia, their paper is not about intrinsic reward, so the type of local optima that they get stuck in are quite distinct from the optima created by the intrinsic reward design that the short-sightedness problem refers to.

I am a strong proponent of using tabular simplifications to better understand problems, but in this case, the tabular simplification has not allowed for a sufficiently detailed understanding of short-sightedness. For this reason, I think it is important to look to the function approximation setting. In the function approximation setting, shortsightedness isn't clearly a problem. Your results in Table 2 suggest that RND doesn't suffer from short-sightedness in the same way that the tabular count-based method does—it does not spread out between the corridors evenly, which seems to be the behaviour you are looking for, but it doesn't get the short-sightedness effect either.

**The pseudo-mathematical analysis in the appendix seems seriously oversimplified and flawed.**
When I said that it was over-simplified, I meant that the assumptions that were made are inappropriate. I am not looking for more complex analysis; I am looking for analysis of an equally simple problem but one that is correct. The primary problem with the analysis is that it sort of stops in the middle. There is a great deal of set-up, and then the final point is simply a qualitative statement that if the growth of a quantity is proportional to how big it is, then the quantity that starts out larger will continue to get larger. The differential equations are not needed and not even used to make this point.

1. While it is true that you do not state that the second-longest corridor will obtain the second-most exploration, I would like to understand what makes the choice of the second corridor to explore different from the choice of the first corridor to explore. From what you've written, it sounds like the agent is going to become obsessed with the longest corridor, and continue exploring it until it is "exhausted." If the first corridor is never exhausted, why are the remaining three corridors not explored roughly the same amount? If the first corridor is exhausted, why isn't the next corridor that the agent becomes obsessed with the second longest, since it is now the longest corridor that is not exhausted, making it a similar choice to the initial one. In this case, wouldn't the second-longest corridor receive the second-most exploration? While I understand that because the action is sampled from the distribution output by the policy network instead of taking max, the "exhausting" would not necessarily happen neatly, but it still seems that the Q-value of the second-longest corridor should become the second-largest, again resulting the second corridor being sampled the second most.

1.3 I am not disputing that this could be thought of as getting stuck in local minima, but I believe that the idea that "alternately feature lower visitation counts" is a completely different problem from the long corridor preference (which may actually have higher visitation counts, but is more preferable due to accumulation).

2.5 Since you do not know why your method outperforms the baselines, it is inappropriate to say that BeBold mitigates the detachment problem.

2.6 I'm sorry that I don't understand this response. Can you explain the relationship between reward clipping and dedication?

3.3 The response does not address my concern. The primary problem is that the way detachment and derailment are described does not seem to be accurate. The example, "in which the agent gets trapped into one (long) corridor and fails to try other choices," partially addresses the problem of detachment, but does not appear to have any relevance to derailment. However, I am also of the opinion that for the authors to adequately support claims about these concepts, they must be described in enough detail for the reader to understand.

4.2 I understand that a difference between RIDE and BeBold is this use of an embedding network, and I can see why you might hypothesize that it might affect performance, but again, I don't think there is any evidence given to show that this is the reason that BeBold outperforms RIDE, so it should be stated appropriately.

6.1 While I am familiar with RND, the problem I am trying to point out here is that the terms appear in the paper prior to any explanation of what they are, which severely hinders readability.

**Additional Feedback (Here to help, not necessarily part of decision assessment)**

I recommend using the terms _target_ network and _predictor_ network for the two networks involved in RND, rather than teacher and student. Since target and predictor are used in the majority of the paper by Burda et al., they're more memorable for readers of the paper, so I think it will be less confusing.

New Typos:
"The results is averaged" → The results are averaged (p. 5)

---

> ### Author Response · Authors · 2020-11-12
> **Author Response for R3 (1/3)**
>
> We thank R3 for all the comments. Please also refer to the common questions above for the answer to the remaining questions
>
> **It is not clear to what extent short-sightedness is a serious issue and how much it affects exploration.**
>
> **A**: The problem is discussed in other literature like Go-Explore [5], First Return then Explore [6], DTSIL [7]. We also conduct experiments combined with analysis on long corridors and MiniGrid. All results are showing that count-based exploration could suffer from this problem.
>
> **The experiments seem to contradict the authors’ reasoning and cast doubt on their analysis.**
>
> **A**: We guess R3 is concerned that the second-longest corridor didn’t receive the second-largest amount of exploration. Note that in neither Sec 4 nor the appendix, do we imply that the second-longest corridor receives the second-largest amount of exploration. What we argue is that the agent could be obsessed with the longest corridor and spends a lot of time there, which is justified by the first row of Table 1. For the second-longest corridor, the situation can be complicated, in particular, if there are more than two corridors.
>
> Note that due to the random initialization of the network, it is not necessarily true that the longest corridor will be visited the most often. Table 1 (row 3) shows that. We leave a more thorough empirical analysis (multiple trials with standard deviation) for the corridor cases in the next revision of the paper.
>
>
> **The pseudo-mathematical analysis in the appendix seems seriously oversimplified and flawed.**
>
> **A**: In this paper, we are mainly focused on proposing a novel criterion and showing its empirical performance. The primary goal of mathematical analysis in the appendix is to provide an understanding of what could happen in a very simple case. Analysis involving complex neural network models, techniques like epsilon-greedy, prioritized experience replay is beyond the scope of this section. We have empirically shown that BeBold works very well even combined with these techniques.
>
> **Q 1.1**: It is implied that the second-longest corridor will receive the second-largest amount of exploration, but this is not true in the results shown in Table 1.
>
> **A**: Please see the second question above.
>
> **Q 1.2**: The assumption in the appendix "the probability to take action left or right is proportional to the accumulated reward" is not generally true.
>
> **A**: The probability to take action left or right is proportional to the accumulated reward" (p. 12) This assumption is not the case for epsilon-greedy Q-learning nor Softmax/Boltzmann exploration. => Note that our work is mostly empirical and the analysis in the appendix is an illustrative example to show why a count-based approach could lead to getting stuck in a long corridor. In the analysis we assume that the policy to be taken is $\pi(left|s) = Q(left|s) / (Q(right|s) + Q(left|s))$, which is already a relatively balanced strategy if one side of Q function is larger than the other side. For epsilon-greedy Q learning or Softmax/Boltzmann exploration, the effect of winner-take-all is more severe due to argmax/exponentiation, therefore the count-based approach can be even more biased towards long corridors, strengthening our conclusions. We will include these cases in the next revision.
>
> **Q 1.3**: The description of short-sightedness in the introduction doesn’t match the problem of short-sightedness described in Section 4.
>
> **A**: We use a high-level description in the introduction and a more detailed description of short-sightedness in Section 4. The fact that the agent gets stuck in a long corridor (or alternating between two corridors) can be regarded as one specific form of local minima.
>
> **Q 1.4**: How does BeBold deal with stochastic environments? In such environments, count-based approaches will return to states multiple times to acquire enough data to account for high variance.
>
> **A**: In our MiniGrid experiments, we compare not only with count-based methods but also state-diff (RIDE) and curiosity-driven (ICM). Results show that BeBold is the only method that can get the maximum reward in hard environments. In a stochastic environment, BeBold could also return to states multiple times. The IR defined in BeBold is the difference of inverse visitation count between consecutive states and it is high only when the visitation count for previous states $s_{t}$ is high and that for current state $s_{t+1}$ is low. This indicates that the agent has already visited state $s_{t}$ multiple times. We have also tested BeBold in MonteZuma’s Revenge. Please see Common Questions 1 for detailed results.
>
> (to be continued)

---

> ### Author Response · Authors · 2020-11-12
> **Author Response for R3 (2/3)**
>
> **Q 2.1 & 2.2**: Why not use tabular Q learning in Tables 1 and 2? Tables 1 and 2 do not appear to include any measure of uncertainty.
>
> **A**: We thank the reviewer for pointing that out. That’s a good point. We’ll add these comparisons and update the paper accordingly in the next revision.
>
> **Q 2.3**: Did BeBold solve all of the environments or only the 12 "most challenging ones"
>
> **A**: After the submission of the paper, we also tested BeBold on all static environments in MiniGrid and it solves all of them. By the time we wrote the paper, it solved 12 “most challenging ones”.
>
> **Q 2.4**: In Table 2 there is no mention of what 0.2M, 0.5M is.
> **A**: This is the number of total time steps.
>
> **Q 2.5**: Do you have quantitative results showing that BeBold mitigates the detachment problem?
>
> **A**: We have shown by extensive experiments that BeBold outperforms some of the baselines which suffer from the detachment problem (RND). The formal theoretical analysis and quantitative results on how BeBold mitigates this problem is beyond the scope of this paper. We leave this for future work.
>
> **Q 2.6**: Can you provide a little more explanation about why clipping needs to be done to allow for a fair comparison in Sec. 5.2?
>
> **A**: The wording here might be a little inaccurate. The clipping here is the same as Sec. 3, only positive rewards are allowed. Since the long-corridor experiment is a toy experiment we designed to illustrate the dedication behavior of count-based approaches, in order to exclude the effect of reward clipping, we didn’t include that in this section.
>
> **Q 2.7**: There are concerns about the choice to measure the quantity of training data based on the number of episodes rather than the number of time steps.
>
> **A**: We use the number of steps in all our experiments except Tab. 1. In Tab. 1, the number of steps is actually similar (same order of magnitude).
>
> **Q 3.1**: OpenAI Gym supports "most work requires either a manually-designed dense reward" but not that not having it causes most algorithms to fail. Why use this reference in the introduction?
>
> **A**: We thank the reviewer for pointing this out. For this reference, we just want to show that manually-designed dense reward is widely used in the RL environments. However, we think that RL doesn’t work well in sparse settings is widely acknowledged. We’ll update accordingly in the next revision.
>
> **Q 3.3 & 3.4**: Detachment and derailment are written as though they are a single concept and the example doesn't really give a very accurate sense of the definitions of either. The explanation of detachment on page 4 does not seem to accurately capture the meaning as it was defined by Go-Explore.
>
> **A**: Since these two problems are well studied in Go-Explore [5], due to the space limit, we choose to briefly discuss these two problems and refer the interested reader to the Go-Explore paper. Due to the same reason, we didn’t give a formal definition of detachment on page 4 since this problem is well-studied in Go-Explore. We choose to use a two-corridor example in order for the readers to intuitively understand this problem.
>
> **Q 4.1**: What is the meaning of the sentence “another method that relies on good-quality generalization because it uses two stages”?
>
> **A**: We mention “good-quality generalization” specifically for RIDE. In order to calculate the IR in RIDE, we first need to train a state embedding network that generates meaningful embeddings of the state. Only after that can we calculate the IR based on the difference of consecutive states. So the optimization of RIDE is actually 2-stage: 1. Train an embedding network to get meaningful embeddings of the states. 2. Train the RL agent given the state embedding representations. Thus, the success of RIDE heavily relies on learning representations that can generalize to novel states. Otherwise, the distance between novel states and previous ones will be meaningless. We acknowledge that we don’t do any experiments to verify this point so we will tune down the tone a bit in the next revision.
>
> **Q 4.2**: Why reliance on quality generalization is the reason that BeBold performs better than RIDE?
>
> **A**: This is one advantage of BeBold over RIDE: the optimization of BeBold is actually one stage and doesn’t rely on training an embedding network. Another problem with RIDE is as pointed out in Sec. 4, the IR of RIDE doesn’t go to 0 even when sufficient exploration is done.
>
> **Q 4.3**: Why do you compare BeBold to AMIGo rather than other examples that use MiniGrid?
>
> **A**: We also compare BeBold with AMIGo, RIDE, RND, ICM in our experiments in Fig. 3. By the time we wrote this paper, AMIGo was considered as the SoTA. In addition to that, AMIGo uses full observation while BeBold only uses partial observation in our policy network.
>
> (to be continued)

---

> ### Author Response · Authors · 2020-11-12
> **Author Response for R3 (3/3)**
>
> **Q 5**: The paper comments on related works but fails to communicate the connection between some of the other sources (e.g., Go-Explore, frontier-based exploration).
>
> **A**: We thank the reviewer for pointing this out. We will update this in the next revision of the paper.
>
> **Q 6.1**: There is no explanation of how RND is designed has been provided, with the same going for any clues as to why RND gets "trapped".
>
> **A**: Please see the common questions section for how RND is designed. As to why RND gets “trapped”, in Sec. 5.2, we hypothesize that this is due to the short-sightedness problem.
>
> **Q 6.2**: The term trajectory should be defined for your context.
>
> **A**: By trajectory, we mean the (state, action, reward) samples collected by the policy. We will define it more clearly in the next revision.
>
> **References**
>
> [1] Burda, Yuri, et al. "Exploration by random network distillation." arXiv preprint arXiv:1810.12894 (2018).
>
> [2] Raileanu, Roberta, and Tim Rocktäschel. "RIDE: Rewarding Impact-Driven Exploration for Procedurally-Generated Environments." arXiv preprint arXiv:2002.12292 (2020).
>
> [3] Campero, Andres, et al. "Learning with amigo: Adversarially motivated intrinsic goals." arXiv preprint arXiv:2006.12122 (2020).
>
> [4] Ecoffet, Adrien, et al. "Go-explore: a new approach for hard-exploration problems." arXiv preprint arXiv:1901.10995 (2019).
>
> [5] Ecoffet, Adrien, et al. "First return then explore." arXiv preprint arXiv:2004.12919 (2020).
>
> [6] Guo, Yijie, et al. "Self-Imitation Learning via Trajectory-Conditioned Policy for Hard-Exploration Tasks." arXiv (2019): arXiv-1907.

---

### Official Review · AnonReviewer5 · 2020-11-06
**Simple approach, more extensive experiments on other domains will be more interesting and convincing**

**Rating:** 5
**Confidence:** 4

**Review:**

Summary: This paper focuses on exploration with intrinsic rewards and proposes to use the difference of inverse visitation count as the intrinsic rewards. The method outperforms some existing baselines with intrinsic rewards in the procedurally-generated tasks (MiniGrid and NetHack).

Clarity: This paper introduces the method and discusses its advantage over the previous work clearly, but some small points are vague and need more clarification (please see the section 'Cons' for the details.)

Originality: As far as I know, the proposed formulation of intrinsic reward is novel, though it is closely related to prior works. For example, in "NEVER GIVE UP: LEARNING DIRECTED EXPLORATION STRATEGIES", the intrinsic reward is also a combination of visitation count in the current episode and lifelong prediction error.

Significance: The proposed idea itself is interesting, and it has shown SoTA results in some challenging tasks. This work will be more significant if there are positive results on other hard exploration tasks (not only procedurally generated tasks), e.g. Montezuma's Revenge.

Pros:
* The proposed method is simple and effective in some challenging procedurally-generated tasks.
* This paper discusses the weakness of the previous work and explains why the proposed method helps address these issues.
* This paper visualizes the change of visitation count during training and thus explicitly demonstrates the behavior of the learning policy.
* This paper conducts an ablative study to investigate the role of different components in the proposed intrinsic reward.

Cons:
* In section 3, "inverse visitation counts as prediction difference". Could you please clarify why the prediction error in RND method can approximate the inverse visitation count? This unclear point seemingly weakens the ground of the proposed method.
* In Figure 3, are some baselines methods missing in some tasks. For example, on "Medium: KeyCorridorS5R3", are there curves for the baselines?
* In section 4, the authors discuss the weakness of the count-based exploration method: detachment. Go-explore tries to solve the detachment problem and shows super good performance on Atari hard-exploration tasks. It will be quite interesting if the proposed method could also significantly outperform the count-based exploration of the Atari games. Currently, the experiments are all conducted on procedurally-generated tasks. Does the proposed method could also work well on other types of domains?
* In section 1, it is mentioned that the curiosity-driven intrinsic reward suffers from the noisy TV problem. The proposed method will also suffer from this problem or not? In other words, is there any failure case that the proposed method will not work?

---

> ### Author Response · Authors · 2020-11-12
> **Author Response for R5**
>
> We thank R5 for all the comments. Please also refer to the common questions above for the answer to the remaining questions
>
> **Q1.** Are baselines methods missing in some tasks in Fig. 3 (e.g., KeyCorridorS5R3)?
>
> **A1.** The baselines are included but got all zero reward so it cannot be seen. We will provide json files for all results.

---

### Author Response · Authors · 2020-11-12
**Author Response for Common Questions**

We are thankful for the reviewers’ insightful comments. We address common concerns here and will reply to each reviewer separately for their specific comments.

**Contribution.**

[Some summarization of our work here.]

1. BeBold addresses the problem of exploring efficiently for a reinforcement learning agent when the reward function is sparse through intrinsic reward.

2. BeBold presents an interesting and very simple approach to progressively explore at the boundaries of the often visited states.

3. BeBold analyzes some key problems (e.g., short-sightedness, derailment, detachment, etc.) in this domain. We conduct extensive empirical experiments to show that BeBold alleviates these problems to some extent.

4. BeBold achieves SoTA performance on MiniGrid and NetHack. We are currently testing BeBold on MonteZuma’s Revenge and the initial results are also promising (please see common questions).

**Common Questions:**

**What is the performance of BeBold on more complicated/stochastic environments (e.g., MonteZuma’s Revenge)?**

We have since obtained promising results on MonteZuma’s Revenge using a CNN based network, 128 parallel environments and 2 billion frames. Bebold got an external reward of ~10000 while RND only got ~6700.
We are also currently investigating a RNN based approach using BeBold and the preliminary results already exceed an external reward of ~14000 in 500M steps while RND obtained a reward of ~4000 at 500M steps.

**Does BeBold also use domain knowledge like Go-Explore which currently is the SOTA of MonteZuma’s Revenge?**

In contrast to Go-Explore which leverages domain knowledge (e.g., image downscaling and coordinate information) our approach is more general and does not use domain knowledge, but only use raw pixels as inputs.

**What’s the connection between prediction error between the teacher and student network and the inverse visitation count in Sec. 3?**
**Why can predictor error approximate the inverse visitation count?**

We follow the paradigm of RND paper [1]. As said in Sec 2.1 (discussion on inverse visitation count) and Sec 2.2 (random network distillation), “there is a fixed and randomly initialized target (teacher) network which sets the prediction problem, and a predictor (student) network is trained on data collected by the agent. This process distills a randomly initialized teacher neural network into a trained student one. The prediction error is expected to be higher for novel states dissimilar to the ones the predictor has been trained on.” So this could be an indication of the inverse visitation count.

**Does the predictor error correspond to the 1/#visitation count?**

R3 raises a great question about $1/N$ versus $1/\sqrt{N}$. We appreciate the detailed comments and will revise our draft accordingly. Note that our conceptual analysis still works with $1 / \sqrt{N}$. We leave a more thorough theoretical analysis that distinguishes the difference between $1/N$ and $1/\sqrt{N}$ as the future work.

**Could you provide some more intuition about Reward Clipping in Sec. 3?**

The reason BeBold uses clipping is that we don’t want to give the agent a negative for returning from a novel state to a familiar state. It might to some extent make the agent more optimistic and willing to explore further, rather than giving up by receiving negative rewards in some steps.

**What’re the limitations on BeBold (e.g., Noisy TV problems) [R4, R5]**

We acknowledge limitations exist on BeBold. First, it doesn’t solve noisy TV problems like all other count-based methods. Second, the usage of the hash table for episodic restriction might be hard to generalize to continuous space environments. We plan to leave these for future work.

We will fix all the grammar errors and typos in the paper. Thanks to all reviewers for pointing them out.

**References**

[1] Burda, Yuri, et al. "Exploration by random network distillation." arXiv preprint arXiv:1810.12894 (2018).

[2] Raileanu, Roberta, and Tim Rocktäschel. "RIDE: Rewarding Impact-Driven Exploration for Procedurally-Generated Environments." arXiv preprint arXiv:2002.12292 (2020).

[3] Campero, Andres, et al. "Learning with amigo: Adversarially motivated intrinsic goals." arXiv preprint arXiv:2006.12122 (2020).

[4] Ecoffet, Adrien, et al. "Go-explore: a new approach for hard-exploration problems." arXiv preprint arXiv:1901.10995 (2019).

[5] Ecoffet, Adrien, et al. "First return then explore." arXiv preprint arXiv:2004.12919 (2020).

[6] Guo, Yijie, et al. "Self-Imitation Learning via Trajectory-Conditioned Policy for Hard-Exploration Tasks." arXiv (2019): arXiv-1907.

---

### Decision · Program_Chairs · 2021-01-07
**Final Decision**

**Decision:**

Reject

**Comment:**

The reviewers have mixed views about this paper.

However, it seems to me that the paper is missing some important related work on near-optimal exploration and it is only picking a couple of superficially similar approaches to look at. In particular, the standard benchmarks of Rmax, UCRL and Posterior Sampling do are not mentioned. I encourage the authors to look at such methods closely. They perform exploration byplanning over a sample or set of possible MDPs.

I also want to raise another issue mentioned by the reviewers. The paper focuses extensively on neural networks, however a count-based metric is inherently for the tabular case. Why would a neural network be appropriate in such a setting? (The authors use a hash table because they are using a large discrete space. However, does it makes sense to essentially uniformly randomly cluster states together? Could there be another, better method? How about continuous spaces?)

The algorithm idea is interesting, and the core is given already in (1) as:  'give reward in newly visited states'. However, the algorithm as described is incomplete.  It is OK as a high-level description, but normally we'd require sufficient detail to reimplement the method from scratch.
You should for example specify how this intrinsic reward value is going to be used. Most of the reviewers, including me, could not understand how a student/teacher network would be  combined with (1) to produce the intended exploration. Please try to explain in as much detail as possible your algorithm in order for the reviewers to be able to make an informed decision.